# RelationAdapter: Learning and Transferring Visual Relation with Diffusion Transformers

**Yan Gong**[1]    **Yiren Song**[2]    **Yicheng Li**[1]    **Chenglin Li**[1]    **Yin Zhang**[1*]

[1]Zhejiang University  [2]National University of Singapore

{gongyan, yichengli, chenglinli, zhangyin98}@zju.edu.cn
yiren@nus.edu.sg

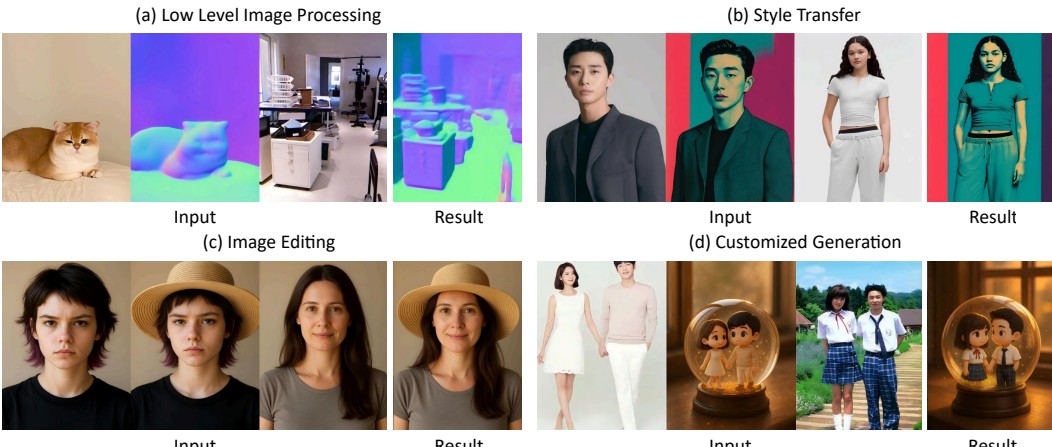

Figure 1: Our framework, RelationAdapter, can effectively perform a variety of image editing tasks by relying on exemplar image pairs and the original image. These tasks include (a) low-level editing, (b) style transfer, (c) image editing, and (d) customized generation.

## Abstract

Inspired by the in-context learning mechanism of large language models (LLMs), a new paradigm of generalizable visual prompt-based image editing is emerging. Existing single-reference methods typically focus on style or appearance adjustments and struggle with non-rigid transformations. To address these limitations, we propose leveraging source-target image pairs to extract and transfer content-aware editing intent to novel query images. To this end, we introduce RelationAdapter, a lightweight module that enables Diffusion Transformer (DiT) based models to effectively capture and apply visual transformations from minimal examples. We also introduce Relation252K, a comprehensive dataset comprising 218 diverse editing tasks, to evaluate model generalization and adaptability in visual prompt-driven scenarios. Experiments on Relation252K show that RelationAdapter significantly improves the model's ability to understand and transfer editing intent, leading to notable gains in generation quality and overall editing performance. Project page: https://github.com/gy8888/RelationAdapter

## 1   Introduction

Humans excel at learning from examples. When presented with just a single pair of images, comprising an original and its edited counterpart, we can intuitively infer the underlying transformation and

---

*Corresponding author: Yin Zhang.

39th Conference on Neural Information Processing Systems (NeurIPS 2025).

apply it to new, unseen instances. This paradigm, known as *edit transfer* or *in-context visual learning* [6, 28, 58, 64], provides an intuitive and data-efficient solution for building flexible visual editing systems. Unlike instruction-based editing methods [18, 25, 61] that rely on textual prompts—where ambiguity and limited expressiveness can hinder precision—image pairs inherently encode rich, implicit visual semantics and transformation logic that are often difficult to articulate in language. By directly observing visual changes, models and users alike can grasp complex edits such as stylistic shifts, object modifications, or lighting adjustments with minimal supervision. As a result, this paradigm offers a highly intuitive and generalizable modality for a wide range of image manipulation tasks, from creative design to personalized photo retouching.

In-context learning-based methods [6, 23, 28, 58, 64] have proven effective in extracting editing intent from image pairs. However, inputting image pairs into the model by concatenating them with the original image leads to several issues, including high memory consumption during inference and degraded performance of text prompts. To address these issues, we aim to develop a dedicated bypass module that can efficiently extract and inject editing intent from example image pairs, thereby facilitating image editing tasks. Nevertheless, building a scalable and general-purpose framework for image-pair-driven editing still presents several fundamental challenges: (1) accurately extracting visual transformation signals from a single image pair, including both semantic modifications (e.g., object appearance, style) and structural changes (e.g., spatial layout, geometry); (2) effectively applying these transformations to novel images while maintaining layout consistency and high visual fidelity; and (3) achieving strong generalization to unseen editing tasks—such as new styles or unseen compositional edits—without requiring retraining.

In this paper, we propose a unified framework composed of modular components that explicitly decouples the extraction of editing intent from the image generation process and enables more interpretable and controllable visual editing.

Our main contributions are summarized as follows:

- First, we propose **RelationAdapter**, the first DiT-based adapter module designed to extract visual transformations from paired images, enabling efficient conditional control for generating high-quality images with limited training samples. A dual-branch adapter is designed to explicitly model and encode visual relationships between the pre-edit and post-edit images. It utilizes a shared vision encoder [40, 65] (e.g., SigLIP) to extract visual features, subsequently injecting these pairwise relational features into the Diffusion Transformer (DiT) [37] backbone to effectively capture and transfer complex edits. As a result, our framework robustly captures transferable edits across semantic, structural, and stylistic dimensions.

- Second, We introduce **In-Context Editor**, a consistency-aware framework for high-fidelity, semantically aligned image editing with strong generalization to unseen tasks. It performs zero-shot image editing by integrating clean condition tokens with noisy query tokens. This mechanism enables the model to effectively align spatial structures and semantic intentions between the input and its edited version. A key innovation introduced in this method is *positional encoding cloning*, which explicitly establishes spatial correspondence by replicating positional encodings from condition tokens to target tokens, thus ensuring precise alignment during the editing process.

- Third, to facilitate robust generalization across a wide range of visual editing scenarios [4, 22, 48], we construct a large-scale dataset comprising **218 diverse editing tasks**. These scenarios span from low-level image processing to high-level semantic modifications, user-customized generation, and style-guided transformations. The dataset consists of **33,274 image pairs**, which we further perform permutation to obtain a total of **251,580 training** instances. This extensive and heterogeneous dataset improves the model's generalization to unseen styles and edits. Furthermore, this dataset provides a unified and scalable foundation for training and evaluating future image-pair editing models.

## 2 Related Work

### 2.1 Diffusion Models

Diffusion models have emerged as a dominant paradigm for high-fidelity image generation [42, 69, 70], image editing[32, 71, 72], video generation [50, 51, 56] and other applications [9, 47, 53, 54]. Foundational works such as Denoising Diffusion Probabilistic Models [20] and Stable Diffusion [42] established the effectiveness of denoising-based iterative generation. Building on this foundation, methods like SDEdit [32] and DreamBooth [43] introduced structure-preserving and personalized editing techniques. Recent advances have shifted from convolutional U-Net backbones to Transformer-based architectures, as exemplified by Diffusion Transformers (DiT) [37, 73] and FLUX [1]. DiT incorporates adaptive normalization and patch-wise attention to enhance global context modeling, while FLUX leverages large-scale training and flow-based objectives for improved sample fidelity and diversity. These developments signal a structural evolution in diffusion model design, paving the way for more controllable and scalable generation.

### 2.2 Controllable Generation

Controllability in diffusion models has attracted increasing attention, with various approaches enabling conditional guidance. ControlNet [68], T2I-Adapter [33], and MasaCtrl [5] inject external conditions—such as edges, poses, or style cues—into pretrained models without altering base weights. These zero-shot or plug-and-play methods offer flexibility in structure-aware generation. In parallel, layout- and skeleton-guided frameworks such as GLIGEN [27] and HumanSD [24] enable high-level spatial control. Fine-tuning-based strategies, including Concept Sliders [15] and Finestyle [66], learn attribute directions or attention maps to enable consistent manipulations. In the era of Diffusion Transformers, some methods concatenate condition tokens with denoised tokens and achieve controllable generation through bidirectional attention mechanisms or causal attention mechanisms [16, 22, 49, 51, 52, 55]. Despite their success, many of these methods rely on fixed condition formats or require significant training overhead [30, 46, 60].

### 2.3 Image Editing

Text-based and visual editing with diffusion models has seen rapid development. Prompt-to-Prompt [18] and InstructPix2Pix [4] allow fine-grained edits using prompt modifications or natural language instructions. Paint by Example [63] and LayerDiffusion [67] exploit visual references and layered generation to perform localized, high-quality edits. Versatile Diffusion [62] supports joint conditioning on text and image modalities, expanding the space of compositional control. Complementary to existing methods that often introduce a substantial number of additional parameters, our proposed RelationAdapter provides a lightweight yet effective solution that leverages DiT's strong pretrained visual representation and structural modeling capacity, enabling few-shot generalization to novel and complex editing tasks. By injecting learned edit intent into DiT's attention layers, our method supports fine-grained structural control and robust style preservation.

## 3 Methods

In this section, we present the overall architecture of our proposed methods in Section 3.1. Next, Section 3.2 outlines our RelationAdapter module, which serves as a visual prompt mechanism to effectively guide image generation. We then integrate the In-Context Editor module (Section 3.3) by incorporating the Low-Rank Adaptation (LoRA) [21] fine-tuning technique into our framework. Finally, Section 3.4 presents a novel dataset of 218 in-context image editing tasks to support a comprehensive evaluation and future research.

### 3.1 Overall Architecture

As shown in Figure 2, our method consists of two main modules:

**RelationAdapter.** RelationAdapter is a lightweight module built on the DiT architecture. By embedding a novel attention processor in each DiT block, it captures visual transformations and

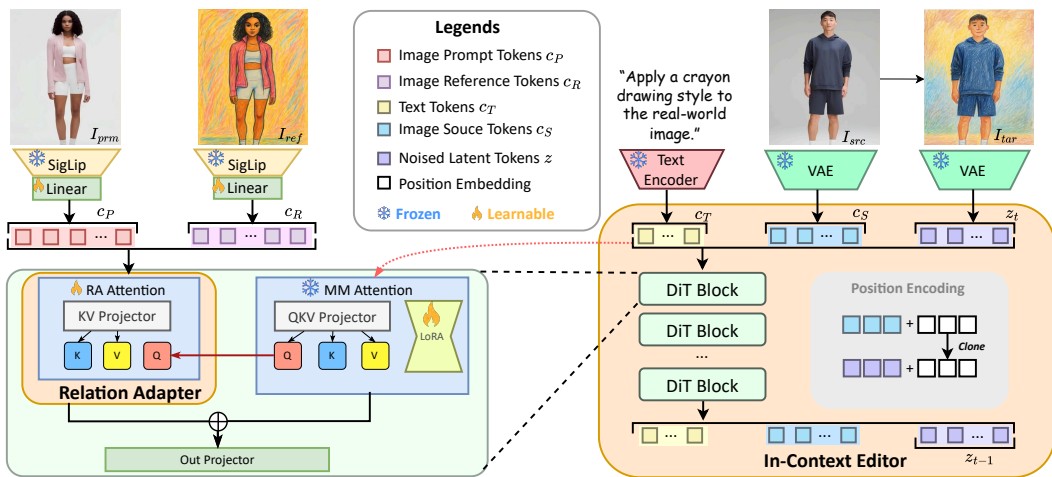

Figure 2: **The overall architecture and training paradigm of RelationAdapter.** We employ the RelationAdapter to decouple inputs by injecting visual prompt features into the MMAttention module to control the generation process. Meanwhile, a high-rank LoRA is used to train the In-Context Editor on a large-scale dataset. During inference, the In-Context Editor encodes the source image into conditional tokens, concatenates them with noise-added latent tokens, and directs the generation via the MMAttention module.

injects them into the hidden states. This enhances the model's relational reasoning over image pairs without modifying the core DiT structure.

**In-Context Editor.** In-Context Editor frames image editing as a conditional generation task during training. It jointly encodes the images and textual description, enabling bidirectional attention between the denoising and input branches. This facilitates precise, instruction-driven editing while preserving the pre-trained DiT architecture for compatibility and efficiency.

## 3.2 RelationAdapter

Our method can be formulated as a function that maps a set of multimodal inputs, namely, a visual prompt image pair $(I_{\text{prm}}, I_{\text{ref}})$, a source image $I_{\text{src}}$, and a textual prompt $T_{\text{prm}}$ to a post-edited image as a target image $I_{\text{tar}}$:

$$I_{\text{tar}} \equiv \mathcal{E}(I_{\text{prm}}, I_{\text{ref}}, I_{\text{src}}, T_{\text{prm}}) \equiv \mathcal{D}\left(\mathcal{R}(I_{\text{prm}}, I_{\text{ref}}), I_{\text{src}}, T_{\text{prm}}\right) \tag{1}$$

where $\mathcal{D}$ denotes the Diffusion Transformer, and $\mathcal{R}$ refers to the RelationAdapter module integrated into the Transformer encoder blocks of the DiT architecture.

**Image Encoder.** Most personalized generation methods use CLIP [40] as an image encoder, but its limited ability to preserve fine-grained visual details hinders high-fidelity customization. To overcome this, we adopt the *SigLIP-SO400M-Patch14-384* [65] model for its superior semantic fidelity in extracting visual prompt features from paired visual prompts $I_{\text{prm}}$ and $I_{\text{ref}}$. Let $\mathbf{c}_P$ and $\mathbf{c}_R$ denote the representations of the sequence of features of $I_{\text{prm}}$ and $I_{\text{ref}}$, respectively. The visual prompt representation $\mathbf{c}_V$ is constructed by concatenating $\mathbf{c}_P$ and $\mathbf{c}_R$.

**Revisiting Visual Prompt Integration.** To enhance the representational flexibility of the DiT based model, we revisit the current mainstream image prompt based approaches (e.g., FLUX.1 Redux [3], which directly appends visual features to the output of the T5 encoder [31]).

Given the visual prompt features $\mathbf{c}_V$ and the backbone DiT input features $\mathbf{c}_B$, FLUX.1 Redux applies a bidirectional self-attention mechanism over the concatenated feature sequence. The resulting attention output $\mathbf{Z}'$ is computed as:

$$\mathbf{Z}' = \text{Attention}(\mathbf{Q}, \mathbf{K}, \mathbf{V}) = \text{Softmax}\left(\frac{\mathbf{Q}\mathbf{K}^{\top}}{\sqrt{d}}\right)\mathbf{V} \tag{2}$$

$$\mathbf{Q} = \mathbf{c}_{B,V}\mathbf{W}_q, \quad \mathbf{K} = \mathbf{c}_{B,V}\mathbf{W}_k, \quad \mathbf{V} = \mathbf{c}_{B,V}\mathbf{W}_v \tag{3}$$

and $\mathbf{c}_{B,V}$ denotes the concatenation of backbone DiT input features $\mathbf{c}_B$ and visual features $\mathbf{c}_V$.

**Decoupled Attention Injection.** A key limitation of current approaches is that visual prompts $\mathbf{c}_V$ are typically much longer than textual prompts $\mathbf{c}_T$, which can weaken or even nullify text-based guidance. We design a separate key-value (KV) attention projection mechanism, $\mathbf{W}'_k$ and $\mathbf{W}'_v$, for the *visual prompts*. Crucially, the cross-attention layer for visual prompts shares the same query $\mathbf{Q}$ with the backbone DiT branch:

$$\mathbf{Z}_V = \text{Attention}(\mathbf{Q}, \mathbf{K}', \mathbf{V}') = \text{Softmax}\left(\frac{\mathbf{Q}(\mathbf{K}')^\top}{\sqrt{d}}\right)\mathbf{V}' \tag{4}$$

$$\mathbf{Q} = \mathbf{c}_B\mathbf{W}_q, \quad \mathbf{K}' = \mathbf{c}_V\mathbf{W}'_k, \quad \mathbf{V}' = \mathbf{c}_V\mathbf{W}'_v \tag{5}$$

Then, we fuse the visual attention output $\mathbf{Z}_V$ (from the RelationAdapter) with the original DiT attention output $\mathbf{Z}_B$ before passing it to the Output Projection module:

$$\mathbf{Z}_{\text{new}} = \mathbf{Z}_B + \alpha \cdot \mathbf{Z}_V \tag{6}$$

where $\alpha$ is a tunable scalar coefficient that controls the influence of visual prompt attention.

### 3.3 In-Context Editor

In-Context Editor builds upon a DiT-based pretrained architecture, extending it into a robust in-context image editing framework. Both the source image $I_{\text{src}}$ and the target image $I_{\text{tar}}$ are encoded into latent representations, $c_S$ and $z$ respectively, via a Variational Autoencoder (VAE) [26]. After cloning positional encodings, the latent tokens are concatenated along the sequence dimension to enable Multi-modal Attention [36], formulated as:

$$\text{MMA}\left([z; c_S; c_T]\right) = \text{softmax}\left(\frac{QK^\top}{\sqrt{d}}\right)V \tag{7}$$

Here, $Z = [z; c_S; c_T]$ denotes the concatenation of noisy latent tokens $z$, source image tokens $c_S$, and text tokens $c_T$, where $z$ is obtained by adding noises to target image tokens.

**Position Encoding Cloning.** Conventional conditional image editing models often struggle with pixel-level misalignment between source and target images, leading to structural distortions. To address this, we propose a *Position Encoding Cloning* strategy that explicitly embeds latent spatial correspondences into the generative process. Specifically, we enforce alignment between the positional encodings of the source condition representation $c_S$ and the noise variable $z$, establishing a consistent pixel-wise coordinate mapping throughout the diffusion process. By sharing positional encodings across key components, our approach provides robust spatial guidance, mitigating artifacts such as ghosting and misplacement. This enables the DiT to more effectively learn fine-grained correspondences, resulting in improved editing fidelity and greater theoretical consistency.

**LoRA Fine-Tuning.** To enhance the editing capabilities and adaptability of our framework to diverse data, we constructed a context learning-formatted editing dataset comprising 251,580 samples (see Section 3.4). We then applied LoRA fine-tuning to the DiT module for parameter-efficient adaptation. Specifically, we employed high-rank LoRA by freezing the pre-trained weights $W_0$ and injecting trainable low-rank matrices $A \in \mathbb{R}^{r \times k}$ and $B \in \mathbb{R}^{d \times r}$ into each model layer.

**Noise-Free Paradigm for Conditional Image Features.** Existing In-Context Editor frameworks concatenate the latent representations of source and target images as input to a step-wise denoising process. However, this often disrupts the source features, causing detail loss and reduced pixel fidelity. To address this, we propose a noise-free paradigm that preserves the source features $c_S$ from $I_{\text{src}}$ throughout all denoising stages. By maintaining these features in a clean state, we provide a stable and accurate reference for generating the target image $I_{\text{tar}}$. Combined with position encoding cloning and a Multi-scale Modulation Attention (MMA) mechanism, this design enables precise, localized edits while minimizing unintended modifications.

### 3.4 Relation252K Dataset

We curate a large-scale image editing dataset encompassing **218** diverse tasks, categorized into four main groups based on functional characteristics: **Low-Level Image Processing**, **Image Style Transfer**, **Image Editing**, and **Customized Generation**. The dataset contains **33,274** images and **251,580** editing samples generated through image pair permutations. Figure 3 provides an overview of four task categories. **We open-source the full dataset** to encourage widespread usage and further research in this field.

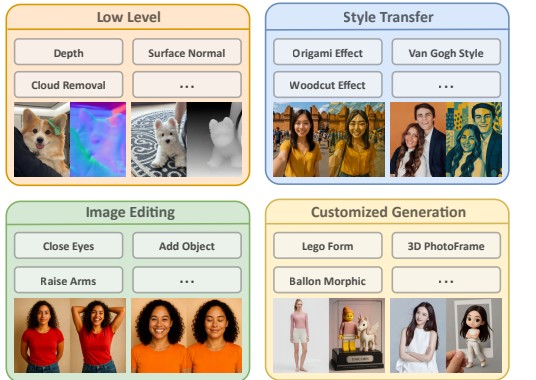

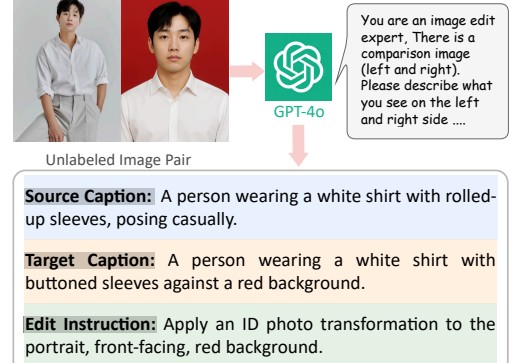

Figure 3: **Overview of four main task categories in our dataset.** Each block lists representative sub-tasks (with ellipses indicating more), along with image-pair examples.

Figure 4: **Overview of the annotation pipeline using GPT-4o.** GPT-4o generates a set of *source caption*, *target caption*, and *edit instruction* describing the transformation from $I_{src}$ to $I_{tar}$.

**Automatic Image Pairs Generation.** We introduce a semi-automated pipeline for constructing a high-quality dataset. A custom script interfaces with a Discord bot to send `/imagine` commands to MidJourney, generating high-fidelity images. Also using the GPT-4o [35] multimodal API, we generate context-aware images from original inputs and edits. For low-level tasks, we additionally curate a subset of well-known benchmark datasets[10, 13, 14, 17, 34, 38, 39, 44] through manual collection to ensure coverage of classic image processing scenarios. Furthermore, part of our original dataset is derived from several existing facial and human image collections, including [11, 12, 29, 45]. To improve annotation efficiency and scalability, we leverage the multimodal capabilities of GPT-4o to automatically generate image captions and editing instructions. Specifically, we concatenate the source image ($I_{src}$) and the corresponding edited image ($I_{tar}$) as a joint input to the GPT-4o API. A structured prompt guides the model to produce three outputs: (1) a concise description of $I_{src}$; (2) a concise description of $I_{tar}$; and (3) a human-readable editing instruction describing the transformation from $I_{src}$ to $I_{tar}$. An example illustrating the pipeline is shown in Figure 4. To conform with the model's input specification, image pairs are sampled and arranged via rotational permutation, with up to 2,000 instances selected per task to ensure distributional balance. In each sample, the upper half is used as visual context for the RelationAdapter, and the lower half is input to the In-Context Editor module. Directional editing instruction ($I_{src} \rightarrow I_{tar}$) are provided solely as text prompt, without detailed content descriptions.

## 4 Experiments

### 4.1 Settings

We initialize our model with FLUX.1-dev [2] within the DiT architecture in training. To reduce computational overhead while retaining the pretrained model's generalization, we fine-tune the In-Context Editor using LoRA, with a rank of 128. Training spans 100,000 iterations on 4 H20 GPUs, with an accumulated batch size of 4. We use the AdamW optimizer and bfloat16 mixed-precision training, with an initial learning rate of $1 \times 10^{-4}$. The total number of trainable parameters is 1,569.76 million. Training takes 48 hours and consumes $\sim$74 GB of GPU memory. At inference, the model requires $\sim$40 GB of GPU memory on a single H20 GPU. The RelationAdapter employs a dual-branch SigLIP visual encoder, where each branch independently processes one image from the input pair and outputs a 128-dimensional feature token via a two-layer linear projection network. The attention

fusion coefficient $\alpha$ is fixed to 1. To balance computational efficiency, input images are resized, maintaining their aspect ratio, such that the longer side does not exceed 512 pixels prior to encoding.

## 4.2 Benchmark

We selected 2.6% of the dataset (6,540 samples) as a benchmark subset, covering a diverse range of 218 tasks. Among these, 6,240 samples correspond to tasks seen during training, while 300 represent unseen tasks used to evaluate the model's generalization capability.

## 4.3 Baseline Methods

To assess the performance of our method, we compare it against two baselines: Edit Transfer [6] and VisualCloze [28]. Both baselines follow an in-context learning setup and are evaluated within the shared training task space to ensure a fair comparison, using the official implementation and recommended hyperparameters to ensure reproducibility.

## 4.4 Evaluation Metrics

We evaluate model performance using five key metrics: **Mean Squared Error (MSE)**, **CLIP-based Image-to-Image Similarity (CLIP-I)**, **Fréchet Inception Distance (FID)**, **Editing Consistency (GPT-C)**, and **Editing Accuracy (GPT-A)**. MSE [59] quantifies low-level pixel-wise differences between the generated and ground-truth images. To capture perceptual and semantic fidelity, we employ both CLIP-I [41] and FID [19]. CLIP-I measures high-level semantic similarity by computing the cosine distance between CLIP embeddings of generated and reference images, while FID evaluates the overall realism and distributional alignment of generated images with real ones in the feature space of a pretrained Inception network, where a lower value indicates higher visual quality. To further assess editing quality from a human-centered perspective, we leverage GPT-4o to interpret the intended transformation from the prompt image $I_{\mathrm{prm}}$ to the reference image $I_{\mathrm{ref}}$, and evaluate the predictions based on two dimensions: Editing Consistency (GPT-C), which measures alignment with the source image $I_{\mathrm{src}}$, and Editing Accuracy (GPT-A), which assesses how faithfully the generated image reflects the intended edit.

## 4.5 Comparison and Evaluation

**Quantitative Evaluation.** As shown in Table 1, our method consistently outperforms the baselines in MSE, CLIP-I, and FID metrics. Compared to Edit Transfer, our model achieves a significantly lower MSE (0.020 vs. 0.043), a higher CLIP-I score (0.905 vs. 0.827), and a reduced FID (4.201 vs. 4.908), indicating better pixel-level accuracy, semantic consistency, and overall visual quality. Similarly, when compared with VisualCloze, our method achieves a notable improvement, reducing the MSE from 0.049 to 0.025, boosting CLIP-I from 0.802 to 0.894, and lowering FID from 7.218 to 4.801. These results demonstrate the effectiveness of our approach in producing both visually accurate and semantically meaningful image edits. Our method also consistently outperforms two state-of-the-art baselines in GPT-C and GPT-A metrics.

**Qualitative Evaluation.** As shown in Figure 5, our method demonstrates strong editing consistency and accuracy in both seen and unseen tasks. Notably, in the unseen task of adding glasses to a person, our approach even outperforms Edit Transfer, which was explicitly trained on this task. In contrast, Edit Transfer shows instability in low-level color control (e.g., clothing color degradation). Compared to VisualCloze, our method is less affected by the reference image $I_{\mathrm{ref}}$, especially in tasks like depth prediction and clothes try-on. VisualCloze tends to overly rely on $I_{\mathrm{ref}}$, reducing transfer accuracy, while our method more reliably extracts key editing features, enabling stable transfer. On unseen tasks, VisualCloze often shows inconsistent edits, such as foreground or background shifts. Our method better preserves structural consistency. This may be due to VisualCloze's bidirectional attention causing feature spillover. Although our method retains some original color in style transfer, it produces more coherent edits overall, indicating room to further improve generalization.

## 4.6 Ablation Study

To assess the effectiveness of our proposed RelationAdapter module, we conducted an ablation study by directly concatenating the visual prompt features with the condition tokens $c_S$. For a fair comparison, this baseline was trained for 100,000 steps, identical to RelationAdapter. As shown in

Table 2, our model consistently outperforms the in-context learning baseline across all five evaluation metrics on both seen and unseen tasks. This improvement is attributed to the RelationAdapter, which enhances performance by decoupling visual features and reducing redundancy.

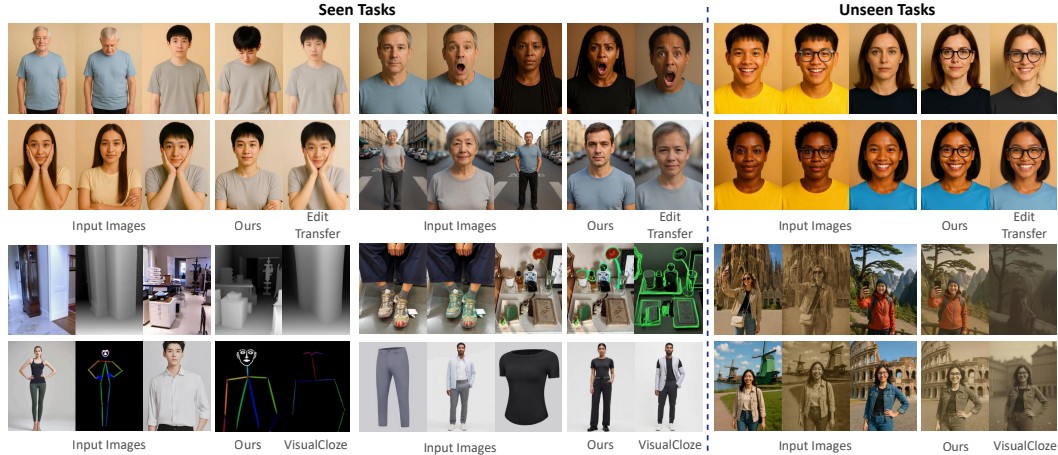

Figure 5: Compared to baselines, RelationAdapter demonstrates outstanding instruction-following ability, image consistency, and editing effectiveness on both seen and unseen tasks.

Table 1: Quantitative Comparison of Baseline Methods Trained on a Common Task (ET: Edit Transfer, VC: VisualCloze). The best results are denoted as Bold.

| Method | $MSE \downarrow$ | $CLIP\text{-}I \uparrow$ | $FID \downarrow$ | $GPT\text{-}C \uparrow$ | $GPT\text{-}A \uparrow$ |
|---|---|---|---|---|---|
| EditTransfer | 0.043 | 0.827 | 4.908 | 4.234 | 3.508 |
| **Ours ∩ ET** | **0.020** | **0.905** | **2.201** | **4.437** | **4.429** |
| VisualCloze | 0.049 | 0.802 | 7.218 | 3.594 | 3.411 |
| **Ours ∩ VC** | **0.025** | **0.894** | **4.801** | **4.084** | **3.918** |

Table 2: Ablation Study on the Effectiveness of the RelationAdapter(RA) in Seen and Unseen Tasks (-S for Seen, -U for Unseen). The best results are denoted as Bold.

| Method | $MSE \downarrow$ | $CLIP\text{-}I \uparrow$ | $FID \downarrow$ | $GPT\text{-}C \uparrow$ | $GPT\text{-}A \uparrow$ |
|---|---|---|---|---|---|
| w/o RA -S | 0.055 | 0.787 | 5.968 | 3.909 | 3.597 |
| **Ours -S** | **0.044** | **0.852** | **5.191** | **4.079** | **4.106** |
| w/o RA -U | 0.061 | 0.778 | 5.571 | 3.840 | 3.566 |
| **Ours -U** | **0.053** | **0.812** | **5.498** | **4.187** | **4.173** |

Although latent-space concatenation (i.e., directly merging four input images before VAE encoding) is effective, it imposes a considerable computational burden during inference. This limitation restricts the resolution of generated images and compromises fine-grained details during inference. In contrast, our lightweight RelationAdapter provides a more efficient alternative, enabling the model to capture and apply the semantic intent of editing instructions with minimal computational cost. Figure 6 demonstrates that our approach yields higher editing accuracy and consistency in both task settings.

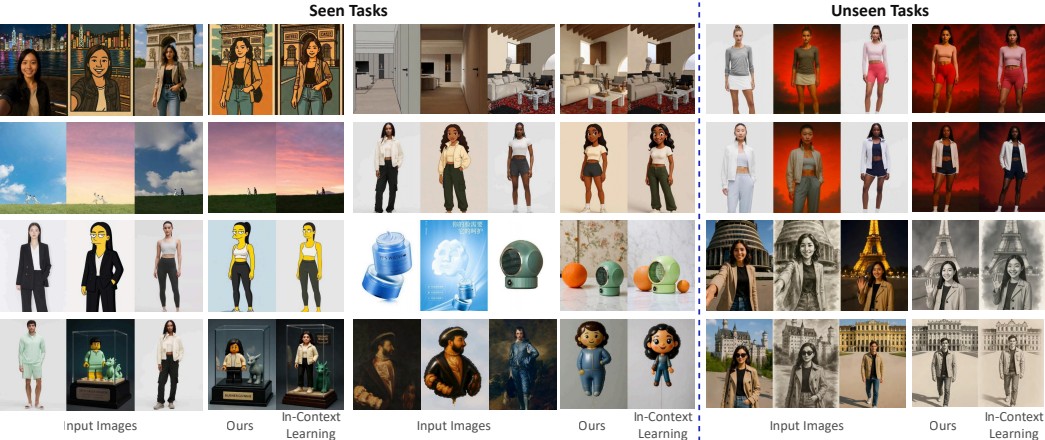

Figure 6: **Ablation study results**. Our strategy shows better editorial consistency.

### 4.7 User Study

We conducted a user study to evaluate our method. Thirty volunteers were recruited to complete assessment questionnaires. In each task, participants were presented with a pair of task prompt images (representing the intended edit), one source image, and two edited results: one generated by our proposed method and the other by a baseline method. For the in-context learning baseline, we used the model variant from our ablation study with the *RelationAdapter* module removed. All images **were randomly sampled** to ensure fairness across tasks. To mitigate potential bias, the order of the two edited images was randomized for each task.

Participants were instructed to interpret the intended transformation from the prompt pair and answer the following three questions:

1. **Edit Accuracy:** Which image better aligns with the editing intent implied by the prompt pair?
2. **Edit Consistency:** Which image better preserves the structure and identity of the source image?
3. **Overall Preference:** Which image do you prefer overall?

The aggregated results of the user study are summarized in Figure 7. When compared with an *in-context learning*-based method, our approach was preferred for tasks included in training in **73.19%** of cases for Edit Accuracy, **80.08%** for Edit Consistency, and **79.58%** for Overall Preference. Even on tasks unseen during training, users still favored our method in **57.67%**, **57.00%**, and **66.33%** of cases, respectively. We also conducted comparisons against other representative baselines. Against *VisualCloze*, our method was preferred in **70.98%** of cases for Edit Accuracy, **72.55%** for Edit Consistency, and **69.22%** for Overall Preference. When compared to *Edit Transfer*, the preference gap widened further, with our method selected in **97.11%** of cases for Edit Accuracy, **78.89%** for Edit Consistency, and **75.78%** for Overall Preference.

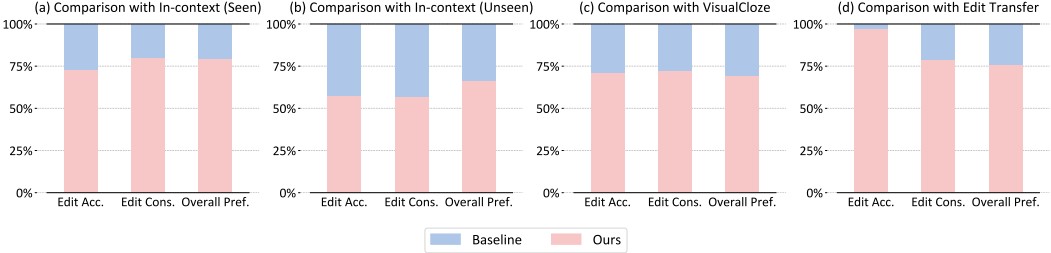

Figure 7: User study results comparing our method with baselines (in-context learning, VisualCloze and Edit Transfer) across evaluation criteria: edit accuracy, edit consistency, and overall preference.

## 5 Discussion

As shown in Figure 8, RelationAdapter demonstrates superior performance in various image editing tasks. This performance can be attributed to the integration of a lightweight module that performs weighted fusion with attention, leading to more precise edits. Notably, this suggests that leveraging visual prompt can be effectively decoupled from conditional generation through attention fusion, without the need for full bidirectional self-attention. This finding reveals a promising direction for designing more efficient and scalable editing models.

Table 3: Quantitative comparison of evaluation metrics (mean $\pm$ std) across four image generation tasks. Best results are shown in bold.

| Tasks | MSE $\downarrow$ | CLIP-I $\uparrow$ | GPT-C $\uparrow$ | GPT-A $\uparrow$ |
|---|---|---|---|---|
| Low-Level (n=32) | **0.028** $\pm$ 0.038 | **0.885** $\pm$ 0.067 | 3.943 $\pm$ 0.383 | 3.822 $\pm$ 0.406 |
| Style Transfer (n=84) | 0.051 $\pm$ 0.032 | 0.846 $\pm$ 0.036 | **4.077** $\pm$ 0.198 | **4.246** $\pm$ 0.285 |
| Image Editing (n=63) | 0.031 $\pm$ 0.023 | 0.861 $\pm$ 0.055 | 4.173 $\pm$ 0.229 | 4.100 $\pm$ 0.400 |
| Customized Generation (n=39) | 0.065 $\pm$ 0.048 | 0.816 $\pm$ 0.073 | 4.071 $\pm$ 0.224 | 4.064 $\pm$ 0.313 |

We evaluated RelationAdapter on four classification tasks of varying complexity. As shown in Table 3, it excels in complex tasks like style transfer and customized generation, showing strong semantic

alignment and text-image consistency. In editing tasks, it balances reconstruction and semantics well. While GPT scores slightly drop in low-level tasks, further low-level evaluations (see supplementary materials B.3) provide a more complete assessment.

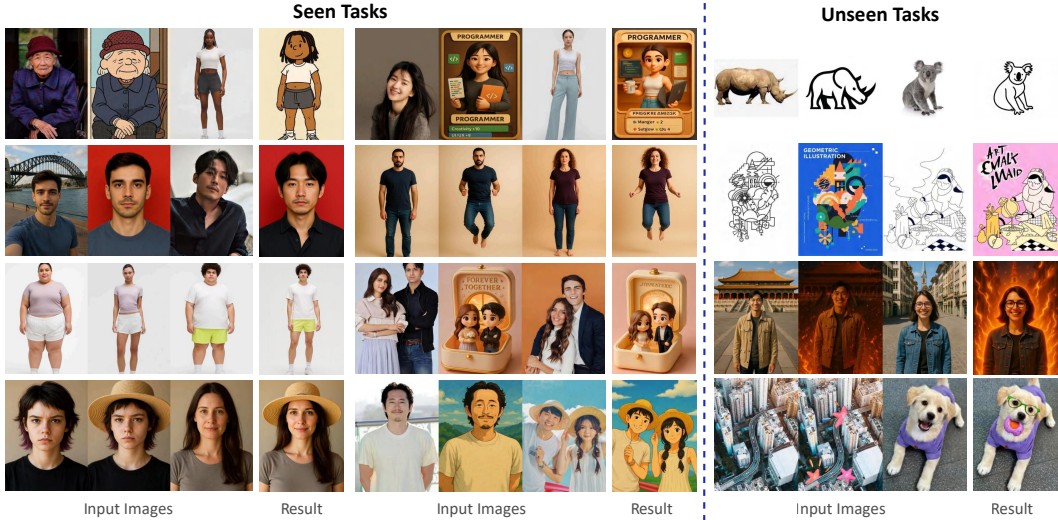

Figure 8: **The generated results of RelationAdapter.** RelationAdapter can understand the transformations in example image editing pairs and apply them to the original image to achieve high-quality image editing. It demonstrates a certain level of generalization capability on unseen tasks.

## 6 Limitation

Although our model performs well across various editing tasks, it sometimes fails to accurately render text details in the generated images. This is a common problem with current Diffusion models. In addition, the model may perform slightly differently on rare or previously unseen tasks, suggesting that it is sensitive to task-specific nuances.

## 7 Conclusion

In this work, we propose RelationAdapter, a novel visual prompt editing framework based on DiT, which strikes a previously unattained balance between efficiency and precision. We begin by revisiting the limitations of existing in-context learning approaches and introduce a decoupled strategy for re-injecting visual prompt features. Leveraging the inherent editing capabilities of DiT, our method enhances both the stability and the generative quality of the model in the in-context learning scenarios. To support our approach, we construct a large-scale dataset comprising 218 visual prompt-based editing tasks. We further introduce two training paradigms-position encoding cloning and a noise-free conditioning scheme for In-Context Editor, which significantly improve the model's editing capability. Extensive experiments validate the effectiveness of our method and demonstrate its superior performance across diverse editing scenarios. We believe this efficient and accurate framework offers new insights into visual prompt-based image editing and lays the groundwork for future research.

## Acknowledgement

This work was supported by the Zhejiang Provincial Natural Science Foundation of China under Grant No. LZ23F020009, the NSFC project (No. 62072399), the Fundamental Research Funds for the Central Universities (No. S20240030), MoE Engineering Research Center of Digital Library, China Research Centre on Data and Knowledge for Engineering Sciences and Technology.

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

# Appendices

The Appendices provide a comprehensive overview of the experimental framework used to develop and evaluate our method. It includes implementation details (Section A), comparisons with baselines (Section B), failure case analysis (Section F), user study design (Section 4.7), and additional results (Section G).

## A  Implementation Details

### A.1  Data Annotation

We leverage the multimodal capabilities of **GPT-4o** to automatically generate image captions and editing instructions. Specifically, we concatenate the source image $I_{\text{src}}$ and the corresponding target image $I_{\text{tar}}$ as a single input to the GPT-4o API. A structured text prompt—illustrated in Figure 9—is provided to guide the model in producing three outputs: **a concise caption for $I_{\text{src}}$; a concise caption for $I_{\text{tar}}$; a human-readable instruction describing the transformation from $I_{\text{src}}$ to $I_{\text{tar}}$.** Notably, the editing instruction is provided solely in textual form, without detailed descriptions of image content.

### A.2  Inference Details

During inference, we set the `guidance_scale` to 3.5, the number of denoising steps to 24, and the attention fusion weight $\alpha$ to 1.0. A fixed random seed of 1000 was used to ensure reproducibility.

## B  Details of Comparisons with Baselines

### B.1  Baseline and Ablation Study Settings

We adopt the official implementations and default configurations for both **VisualCloze** and **Edit Transfer**. During inference, since **VisualCloze** supports layout prompts, we specify the layout as: *"4 images are organized into a grid of 2 rows and 2 columns."* Before concatenating the images into the grid layout, each individual image is resized to a square region with an area of $512 \times 512$ pixels to ensure consistent resolution and layout compatibility. We fix the random seed to 1000 and use the default 30 denoising steps. For **Edit Transfer**, we similarly set the random seed to 1000, while keeping all other parameters at their default values.

In the ablation study, we remove all components related to the **RelationAdapter** module and directly feed the prompt image $I_{\text{prm}}$ and the reference image $I_{\text{ref}}$ into the **In-Context Editor**. Additionally, we apply *Position Encoding Cloning* to each input image to retain spatial correspondence. All other configurations are kept unchanged to ensure fair comparison.

### B.2  Evaluation Details

We leverage the multimodal reasoning capabilities of **GPT-4o** to interpret the intended transformation from the prompt image $I_{\text{prm}}$ to the reference image $I_{\text{ref}}$, and evaluate model predictions from a human-centered perspective along two key dimensions: **Editing Consistency (GPT-C)** and **Editing Accuracy (GPT-A)**.

To facilitate this evaluation, we construct composite inputs consisting of five concatenated images: the prompt image $I_{\text{prm}}$, the reference image $I_{\text{ref}}$ (representing the desired attribute or change), the source image $I_{\text{src}}$, and two generated results $I_{\text{pred}_1}$ and $I_{\text{pred}_2}$. GPT-4o is then prompted to interpret the intended edit and assess each prediction based on the above criteria. The specific text prompt provided to GPT-4o is illustrated in Figure 10.

### B.3  Perceptual Capability Evaluation

We evaluate the model's perceptual capability across a series of low-level image editing tasks, including depth estimation, surface normal prediction, edge detection, and semantic segmentation. We further compare its performance against the current state-of-the-art general-purpose image generation framework, VisualCloze, using multiple evaluation metrics. Detailed results are provided in Tables 4, 5, 6, and 7.

### B.4  Additional Explanation on Baseline Selection

**RelationAdapter** is designed around a unique *before–after* pair formulation, in which the model learns visual transformations directly from exemplar pairs. Among existing approaches, only **Edit Transfer** [6] and **Visual-Cloze** [28] share this paired-context setup, making them the most appropriate baselines for direct comparison.

Table 4: Edge detection performance on the BSDS500 dataset.

| Metric | VisualCloze | Ours |
|---|---|---|
| Precision ↑ | **0.3476** | 0.2266 |
| Recall ↑ | 0.0837 | **0.3134** |
| F1-score ↑ | 0.1227 | **0.2150** |

Table 5: Segmentation performance on the COCO dataset.

| Metric | VisualCloze | Ours |
|---|---|---|
| Pixel Acc. ↑ | **0.7817** | 0.7810 |
| Mean Acc. ↑ | 0.3959 | **0.4722** |
| Mean IoU ↑ | 0.3143 | **0.3642** |

Table 6: Depth estimation ($\delta_1$) on multiple datasets.

| Dataset | VisualCloze | Ours |
|---|---|---|
| BSDS500 | 0.1492 | **0.1833** |
| COCO | 0.1515 | **0.1750** |
| BIPED | 0.2954 | **0.3088** |

Table 7: Surface normal estimation results. Lower error and higher accuracy indicate better performance. *Mean/Median Angular Error* measure deviation from ground truth (°), while *Accuracy@X°* reports the percentage of predictions within X degrees. Best results are highlighted in bold.

| Metric / Dataset | BSDS500 | | COCO | | BIPED | | NYUv2 | |
|---|---|---|---|---|---|---|---|---|
| | VisualCloze | Ours | VisualCloze | Ours | VisualCloze | Ours | VisualCloze | Ours |
| Mean Angular Error (°) | 52.29 | **27.15** | 63.30 | **35.62** | 53.15 | **29.83** | 43.19 | **31.69** |
| Median Angular Error (°) | 49.15 | **24.59** | 61.00 | **32.66** | 51.01 | **28.39** | 37.78 | **28.30** |
| Accuracy ($<11.25°$) | 6.76 | **16.12** | 2.21 | **11.90** | 4.09 | **12.28** | 4.47 | **11.46** |
| Accuracy ($<22.5°$) | 22.10 | **47.89** | 9.59 | **36.30** | 15.06 | **38.57** | 20.74 | **37.33** |
| Accuracy ($<30°$) | 33.52 | **65.88** | 18.13 | **51.68** | 25.56 | **54.87** | 36.38 | **53.75** |

This task formulation distinguishes our method from a broad range of existing image editing and generation frameworks.

Methods such as **Prompt-to-Prompt** [18] and **RF-Edit** [57] operate purely in the text-driven editing paradigm without utilizing visual exemplars, and therefore cannot model transformation relationships between images. **Zero-shot Image Editing** [7] and **OminiControl** [55] focus on reference-conditioned generation, where auxiliary visual signals such as edge maps, depth maps, or segmentation masks are used to guide image synthesis. Their goal is to apply pre-defined visual conditions rather than to learn transferable transformation mappings. **UniReal** [8] addresses a multi-image compositional generation task, e.g., combining the subject from one image with the background of another under mask guidance, which fundamentally differs from exemplar-based transformation learning.

In contrast to the above methods, **RelationAdapter** learns to infer the transformation itself from paired visual exemplars, enabling the transfer of edit intent to unseen content domains. This formulation requires both a source and an edited target image as context, providing explicit supervision for relational transformation understanding.

## B.5 Comparison with Midjourney

Although **Midjourney (MJ)** represents a strong general-purpose image generation system, it does not support pairwise or multi-image conditioning for transformation-based editing. Its interface only distinguishes between a character reference (`-cref`) and a style reference (`-sref`), without the capability to process relational transformations between two visual exemplars. In contrast, RelationAdapter interprets an exemplar pair as a direct demonstration of the intended visual change, which constitutes a distinct learning paradigm.

For completeness, we evaluated Midjourney by assigning the source image of each exemplar pair as the `-cref` and the edited image as the `-sref`, using the following standardized prompt format:

```
[text prompt] --cref <source image> --sref <edited image> --cw 90 --sw 70 --v 6.1
```

Table 8 reports the quantitative comparison on unseen style transfer tasks. Despite Midjourney's strong generative priors, RelationAdapter consistently achieves superior performance across all evaluation metrics, indicating better perceptual consistency and transformation fidelity.

Table 8: Comparison of RelationAdapter and Midjourney (MJ) on unseen style transfer tasks. The best results are denoted in bold.

| Method | MSE ↓ | CLIP-I ↑ | FID ↓ | GPT-C ↑ | GPT-A ↑ |
|---|---|---|---|---|---|
| MJ | 0.107 | 0.681 | 5.836 | 3.285 | 3.200 |
| **Ours** | **0.062** | **0.774** | **5.715** | **4.203** | **4.278** |

## C Advantage over the In-Context Based Variant

We analyze the effectiveness and efficiency gains of **RelationAdapter** over the in-context variant. The in-context method required approximately **77 GB** of GPU memory and **51.5 hours** of training time, whereas our

**RelationAdapter** used around **74 GB** of memory and completed training in about **48 hours**, corresponding to a memory saving of roughly **3 GB** and a **6.8%** reduction in total training time. For inference, editing a single image at a resolution of $1024 \times 1024$ took over **13 seconds** with the in-context method, while RelationAdapter required less than **9 seconds**, achieving a **30.8%** speed-up. These improvements stem from a crucial architectural distinction: the in-context approach concatenates all tokens from exemplar and target contexts, leading to increased attention computation and slower inference, whereas RelationAdapter employs a decoupled attention mechanism that processes and fuses them more efficiently. Moreover, the same mechanism contributes to the observed gains in editing accuracy and consistency by preventing feature contamination between the exemplar pair and target image, enabling more targeted and coherent transformations. Both quantitative evaluations (Table 2) and qualitative visualizations (Figure 6) consistently confirm that RelationAdapter achieves superior perceptual fidelity and structural consistency while offering notable improvements in memory efficiency and processing speed.

## D  Effect of Attention Fusion Coefficient and Visual Encoder Choice

The attention fusion coefficient $\alpha$ controls the relative contribution between the visual prompt attention generated by the RelationAdapter and the base Multi-Modal Attention (MMA) within the Diffusion Transformer. As specified in Section 3.2, we set $\alpha = 1$ during training to maintain a balanced integration between visual guidance and generative consistency, and adopt the same value during inference for training–deployment consistency. To further assess its influence, we varied $\alpha$ across $\{0.5, 1, 2\}$ and report the results in Table 9. The results indicate that maintaining $\alpha = 1$ yields the most stable and optimal generation performance across both seen (–S) and unseen (–U) tasks, while deviating from this setting slightly degrades fidelity and consistency.

Table 9: Effect of adjusting the attention fusion coefficient $\alpha$ on image editing quality. "–S" and "–U" denote seen and unseen tasks, respectively.

| Method | MSE ↓ | CLIP-I ↑ | FID ↓ | GPT-C ↑ | GPT-A ↑ |
|---|---|---|---|---|---|
| $\alpha = 2$–S | 0.044 | 0.827 | 5.564 | 3.855 | 3.536 |
| $\alpha = 1$–S | **0.044** | **0.852** | **5.191** | **4.115** | **4.258** |
| $\alpha = 0.5$–S | 0.050 | 0.832 | 5.895 | 4.099 | 3.591 |
| $\alpha = 2$–U | 0.054 | 0.794 | 5.805 | 3.858 | 3.527 |
| $\alpha = 1$–U | **0.053** | **0.812** | **5.498** | **4.211** | **4.377** |
| $\alpha = 0.5$–U | 0.056 | 0.808 | 5.724 | 4.149 | 3.620 |

## E  Effect of Model Size and Low-Rank Configuration

To assess the impact of model size on performance, we conducted an additional experiment using a lower-rank configuration in the LoRA modules. The number of trainable LoRA parameters decreases from **358.6M** to **44.8M** when reducing the rank from 128 to 16, corresponding to an **87.5%** reduction in trainable parameters. As shown in Table 10, the proposed method remains robust and effective under this lightweight configuration, exhibiting only marginal performance degradation.

Table 10: Comparison between LoRA rank = 16 and the original configuration. "–S" and "–U" denote seen and unseen tasks, respectively.

| Method | MSE ↓ | CLIP-I ↑ | FID ↓ | GPT-C ↑ | GPT-A ↑ |
|---|---|---|---|---|---|
| Ours–S (Rank=16) | 0.048 | 0.828 | 5.757 | 4.035 | 3.607 |
| Ours–S | **0.044** | **0.852** | **5.191** | **4.145** | **4.219** |
| Ours–U (Rank=16) | 0.064 | 0.792 | 5.924 | 4.026 | 3.505 |
| Ours–U | **0.053** | **0.812** | **5.498** | **4.195** | **4.239** |

## F  Failure Cases

Figure 11 illustrates a set of challenging editing tasks. While the model successfully captures edit intentions in several cases, it struggles with fine-grained spatial alignment and the restoration of detailed textual elements. A future solution could involve training on higher-resolution data to better capture spatial nuances.

## G  Additional Results

As shown in Figures 12, 13, and 14, our method demonstrates strong performance across diverse editing tasks, effectively handling spatial transformations and capturing complex semantic modifications with high fidelity.

**Text Prompts**

This is a side-by-side comparison image (left and right).
Please describe what you see on the left and right side respectively,
and provide a transformation or edit instruction from left to right.
Return only a JSON object with the following fields:
1. 'left_image_description'
2. 'right_image_description'
3. 'edit_instruction'
Do not include any other text or explanation.

Figure 9: Structured prompt used for labeling image pairs and extracting transformation instructions.

**Text Prompts**

You are given a composite image with two columns.
The left column contains three images arranged vertically: Left Column: A (original image), A1 (edited version of A), B (another original image).

The right column contains two images: B1 and B2, which are two independently edited versions of image B.

Your task is to independently score B1 and B2 based on two dimensions:

1. Edit Consistency (1–5): How visually consistent is the edited image (B1 or B2) with the original image B? Focus: Are key objects, colors, and structures consistent with the source?

2. Edit Accuracy (1–5): Assess how accurately the editing operation applied to B (to produce B1 or B2) mirrors the transformation seen from A → A1. Focus: Did the editor apply similar changes, in the correct location, with the same degree of modification?

Avoid giving tied scores unless B1 and B2 are truly indistinguishable. Ensure scores reflect nuanced differences in both consistency and accuracy between B1 and B2. Be critical. Reserve scores of 4–5 for highly consistent/accurate edits. Be objective and concise in your assessment.

Return your answer in the following **JSON format**: { "B1": {"consistency":<1-5>, "accuracy":<1-5>}, "B2": {"consistency":<1-5>, "accuracy":<1-5>} }

Figure 10: Evaluation prompt used to assess edit consistency and accuracy between two generated outputs, leveraging GPT-4o for interpretation and scoring.

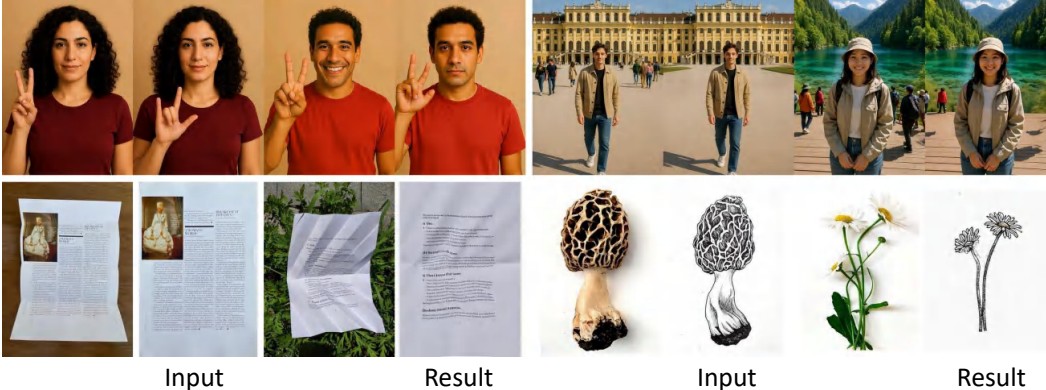

| Input | Result | Input | Result |

Figure 11: Failure cases on gesture editing, background pedestrian removal, document rectification, and image-to-sketch conversion. The model shows partial success with room for improvement.

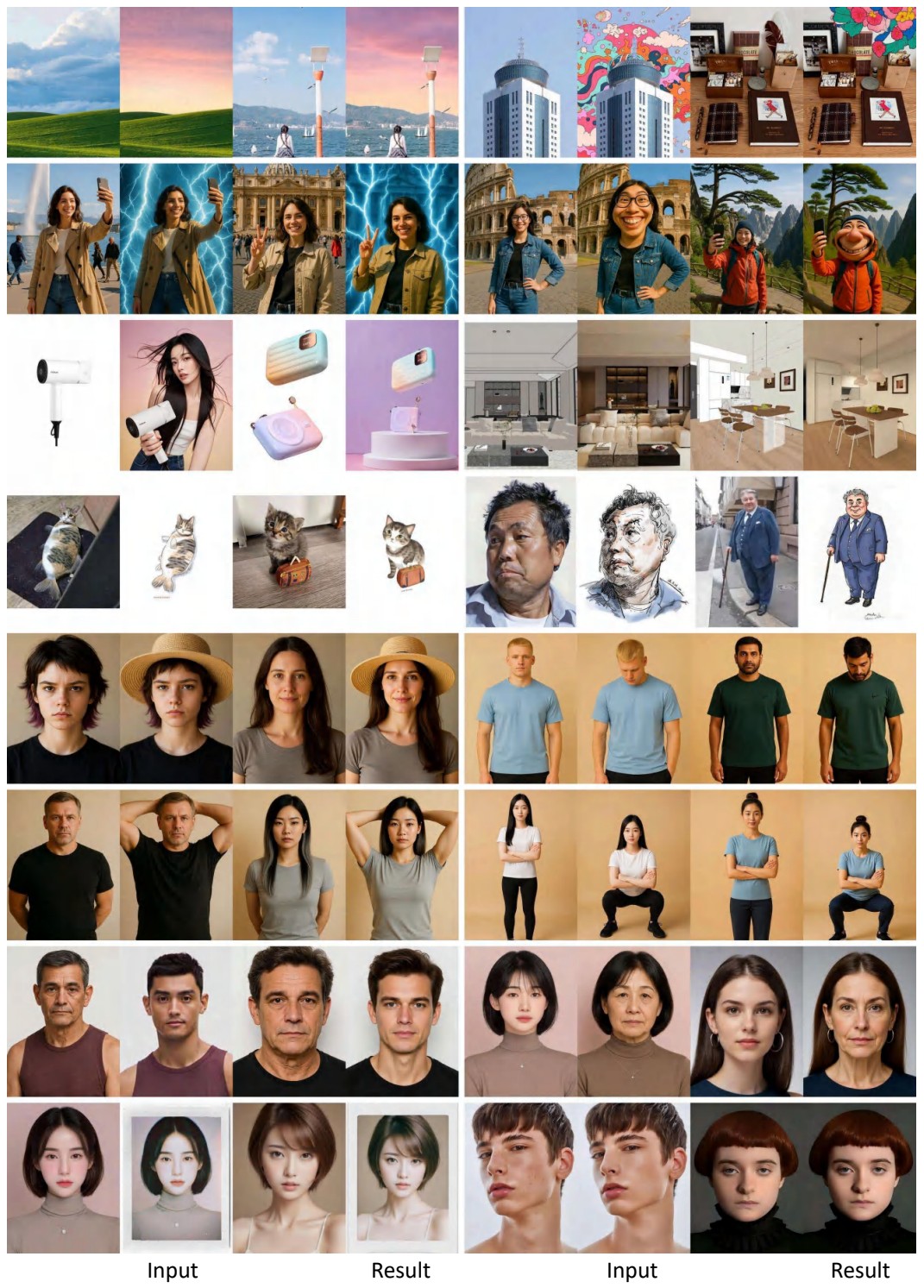

Input      Result      Input      Result

Figure 12: Additional experimental results of RelationAdapter.

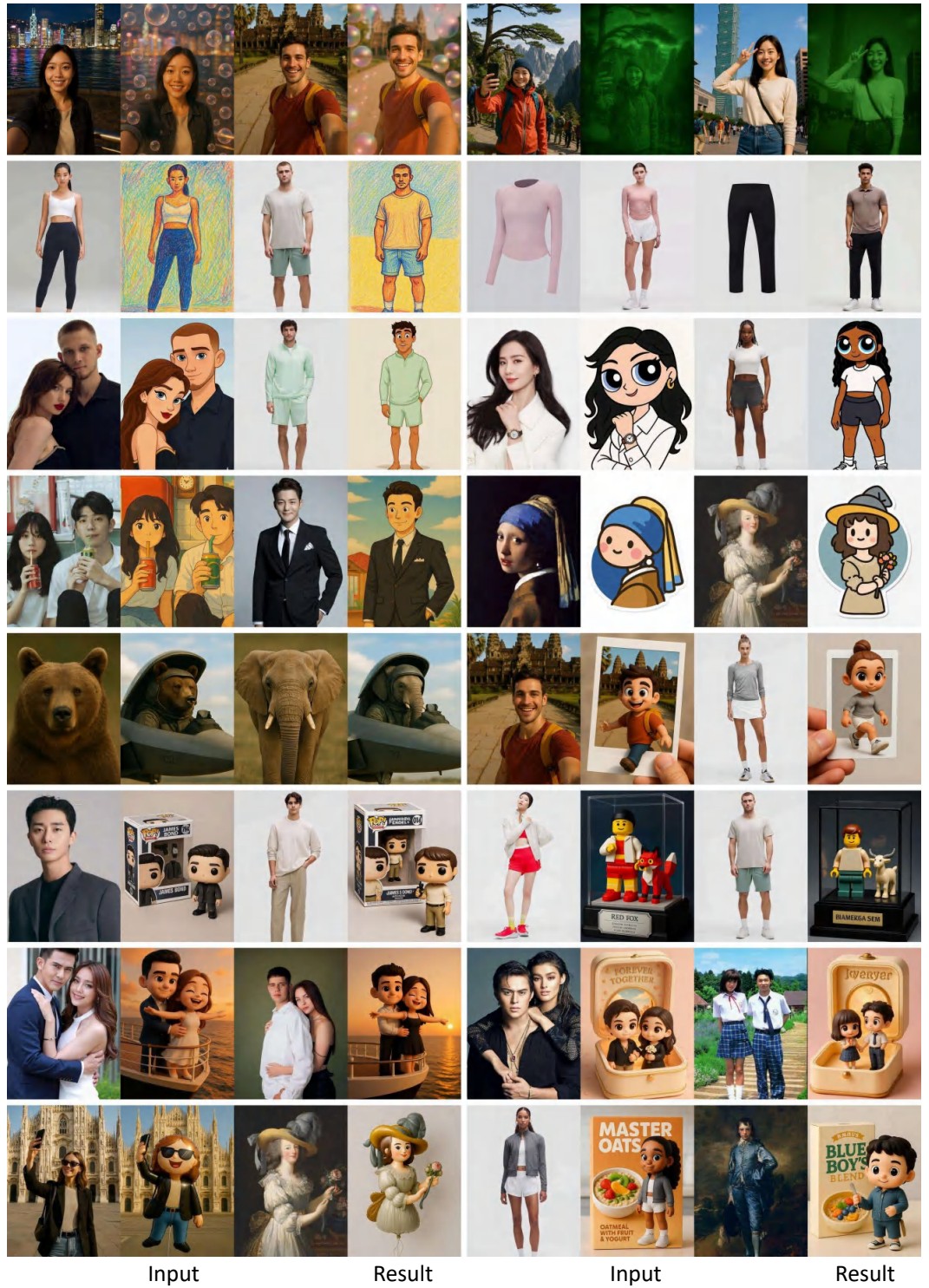

Input        Result        Input        Result

Figure 13: Additional experimental results of RelationAdapter.

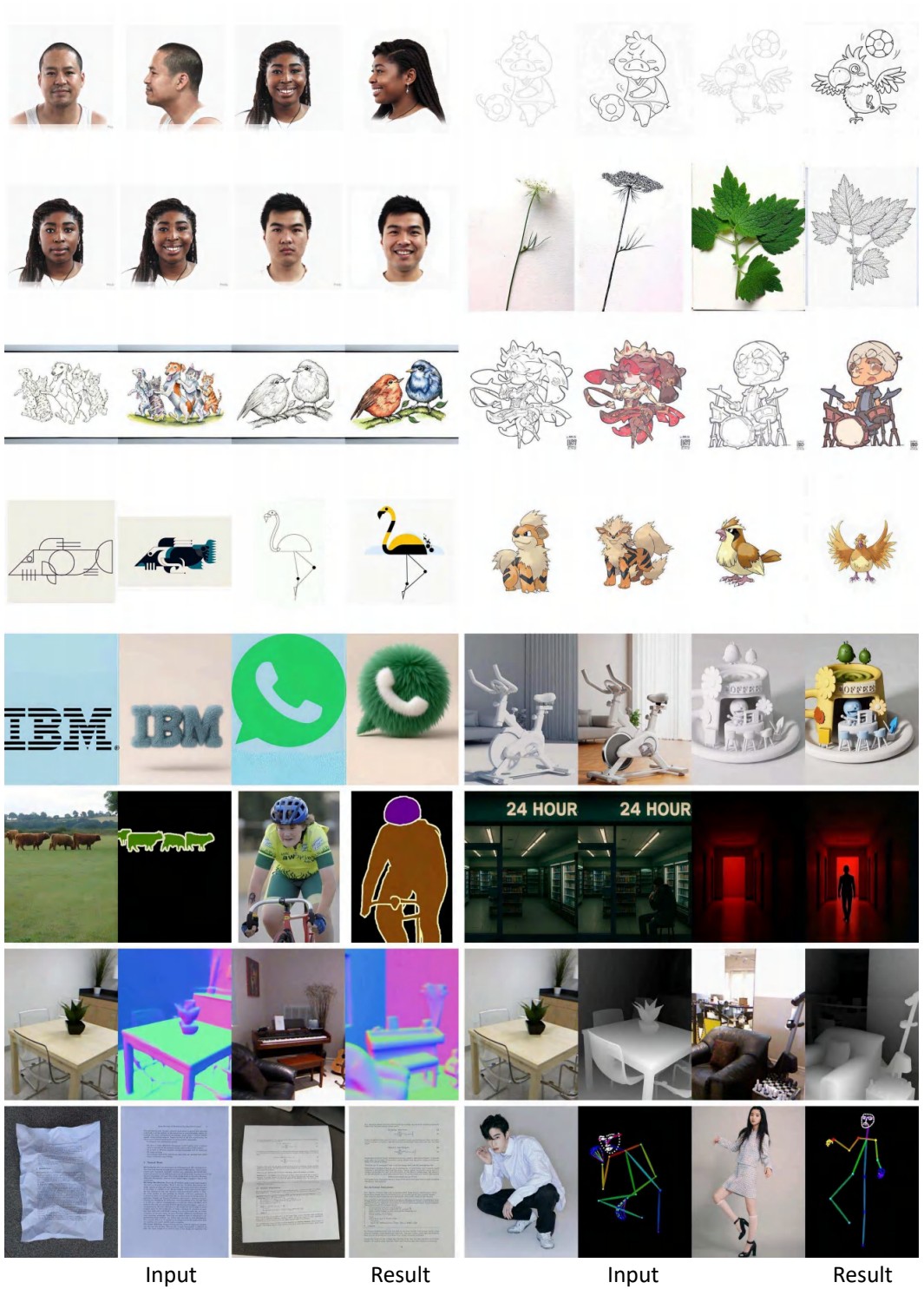

Input        Result        Input        Result

Figure 14: Additional experimental results of RelationAdapter.

