# OpenReview forum: "RelationAdapter: Learning and Transferring Visual Relation with Diffusion Transformers"
_NeurIPS.cc/2025/Conference — NeurIPS 2025 poster_

### Official Review · Reviewer_s3JJ · 2025-07-01

**Clarity:** 3
**Significance:** 2
**Originality:** 2
**Rating:** 3
**Confidence:** 4

**Summary:**

The paper proposes a novel approach for in-context learning of visual transformation from a single source-target image pair and applying it to query images in MMDiT-based text-to-image diffusion models. The authors demonstrate the generalization of the proposed approach on unseen editing tasks. Further, the paper proposes a large-scale visual editing dataset and benchmark with 218 diverse editing scenarios.

**Questions:**

My primary concerns are limited baseline, limited evaluation and missing ablation studies:
1. The authors should include comparisons with atleast two additional baselines including [7] to demonstrate the effectiveness of their approach. Further, the authors should consider comparing [1] using the same LORA Rank for a fair comparison.
2. The authors should evaluate the quality of generated images with FID score.
3. The authors should discuss the effect of parameter $\alpha$ on generation. Further, the authors should provide an experiment to back their claim that SigLIP is a better alternative compared to VAE or CLIP for extracting visual prompt features from paired images.

**Ethical Concerns:**

["NO or VERY MINOR ethics concerns only"]

**Final Justification:**

While my concerns regarding quality of generated images, comparison with baselines under same LORA rank, and motivation for the use of SigLIP have been resolved, my concerns around limited baselines, ablation for parameter , and large number of trainable parameters still remain. 1) Example pair used in the proposed method can be interpreted as visual conditions and text prompt can be used to guide UniReal[7] to generate image from source image following visual relation of the pair (for example, "Generate RES1 from IMG3 by applying the visual relation of IMG2 from IMG1"). 2) It is expected that the  used in training would be the optimal value during inference. However, effect of varying  in the training is unclear and  might not be the optimal value for best performance. 3) Rank 16 does not lead to a significant reduction in the parameter count which reduces from 1.5B to around 1.25B. Hence, I will keep my score.

**Limitations:**

yes

**Quality:**

2

**Strengths And Weaknesses:**

**Strength**

1. The paper proposes a novel adapter for MMDiT-based architecture to extract visual transformation between a source-target image pair and apply transformation to a query image.
2. The authors demonstrate the generalization of the proposed approach on unseen editing tasks along with a user-study demonstrating improved editing preference over baseline.
3. The paper is overall well-written and easy to follow with clear motivation.

**Weakness**

1. Limited baselines: The proposed method is compared with just two baselines, Edit Transfer[1] and VisualCloze[2]. Additional baselines[3,4,5,6,7] should be considered and differences with existing MM-DiT based methods[1,2] should be discussed in related work. Further, comparison with Edit Transfer[1] is not fair as it is trained with much smaller LORA rank 16 while this work uses a high LORA rank of 128.
2. Missing evaluation: The work is missing evaluation comparing the quality of the generated images, of which MSE is not a good indicator, and metrics such as FID should be considered.
3. Missing ablation studies: The work is missing ablation study for attention fusion coefficient $\alpha$ in Eq 6. Further, quantitative comparison should be provided to back claims regarding SigLIP being a better alternative compared to VAE or CLIP for extracting visual prompt features from paired images.
4. While the work claims that the training is parameter efficient, the total number of trainable parameters are 1.5B which is relatively high.


[1] Lan Chen, Qi Mao, Yuchao Gu, and Mike Zheng Shou. Edit transfer: Learning image editing via vision in-context relations.

[2] Zhong-Yu Li, Ruoyi Du, Juncheng Yan, Le Zhuo, Zhen Li, Peng Gao, Zhanyu Ma, and Ming-Ming Cheng. Visualcloze: A universal image generation framework via visual in-context learning.

[3] Amir Hertz, Ron Mokady, Jay Tenenbaum, Kfir Aberman, Yael Pritch, and Daniel Cohen-Or. Prompt-to-prompt image editing with cross attention control. In ICLR, 2023.

[4] Jiangshan Wang, Junfu Pu, Zhongang Qi, Jiayi Guo, Yue Ma, Nisha Huang, Yuxin Chen, Xiu Li, and Ying Shan. Taming rectified flow for inversion and editing.

[5] Xi Chen, Yutong Feng, Mengting Chen, Yiyang Wang,Shilong Zhang, Yu Liu, Yujun Shen, and Hengshuang Zhao. Zero-shot image editing with reference imitation. In NeurIPS, 2025.

[6] Zhenxiong Tan, Songhua Liu, Xingyi Yang, Qiaochu Xue, and Xinchao Wang. Ominicontrol: Minimal and universal control for diffusion transformer.

[7] Chen, Xi, et al. "Unireal: Universal image generation and editing via learning real-world dynamics." Proceedings of the Computer Vision and Pattern Recognition Conference. 2025.

---

> ### Author Rebuttal · Authors · 2025-07-31
>
> ## 1. The authors should include comparisons with at least two additional baselines including [7] to demonstrate the effectiveness of their approach. Further, the authors should consider comparing [1] using the same LORA Rank for a fair comparison.
>
> Thank you for your feedback. Below, we elaborate on the rationale behind our baseline choices:
>
> RelationAdapter requires a complete pair of 'before' and 'after' edit examples to be provided as context, whereas none of the other methods mentioned above rely on such a pair. This formulation is unique to Edit Transfer [1] and VisualCloze [2].
>
> Prompt-to-Prompt [3] and RF-Edit [4] operate entirely in the text-driven editing paradigm. They do not utilize any visual examples and thus lack the capacity to learn visual editing transformations from exemplar pairs. Zero-shot Image Editing [5] and OminiControl [6] focus primarily on reference-conditioned generation, where an auxiliary image—typically a Canny edge map, depth map, segmentation mask, or color scribble—is used to provide fine-grained control signals during image generation. The core idea is to apply a predefined image condition to guide image generation, rather than to learn and transfer a transformation between image pairs. UniReal [7] tackles a different task setup—the combination of multiple conditioning images (e.g., "place the cat from IMG1 onto the bed in IMG2 guided by a mask in IMG3").
>
> Regarding the comment on using the same LoRA rank as [1] for fair comparison, we note that our Relation252K dataset is significantly larger and more diverse (218 tasks) than that of Edit Transfer. Based on our experience, a larger dataset generally requires a higher LoRA rank.
>
> Nevertheless, we have also tested the performance of our model with a LoRA rank of 16 (Table 1). For ease of comparison, Table 2 includes results with LoRA rank 128, showing that our method performs well even with lower ranks.
>
> **Table 1: Comparison of RelationAdapter and ET (Edit Transfer) with LoRA rank=16. The best results are denoted as Bold.**
>
> Method            | MSE ↓    | CLIP-I ↑   | FID ↓    | GPT-C ↑   | GPT-A ↑
> ------------------|----------|----------|----------|-----------|---------
> EditTransfer      | 0.043    | 0.827    | 4.908   | 4.278    | 3.716
> **Ours∩ET (Rank=16)**| **0.021** | **0.868**| **2.892**| **4.483** | **4.533**
>
> **Table 2: Quantitative Comparison of ET (Edit Transfer) Trained on a Common Task. The best results are denoted as Bold.**
>
> Method          | MSE ↓     | CLIP-I ↑   | FID ↓     | GPT-C ↑    | GPT-A ↑
> ----------------|-----------|------------|-----------|------------|-----------
> EditTransfer    | 0.043     | 0.827      | 4.908    | 4.234      | 3.508
> **Ours∩ET (Rank=128)**| **0.020** | **0.905**  | **2.201**| **4.437**  | **4.429**
>
> Note that due to the inherent variability in GPT model scoring, some fluctuations in the GPT-C and GPT-A metrics are normal.
>
> ## 2. The authors should evaluate the quality of generated images with FID score.
>
> We appreciate the reviewer’s valuable suggestion. We have now included FID as a new evaluation metric to further assess the quality of the images produced by RelationAdapter.
>
> As shown in Table 3, our model outperforms the baseline methods in the FID score, indicating a clear improvement in the quality of generated images.
>
> **Table 3: FID Results for Baseline Comparisons (ET: Edit Transfer, VC: VisualCloze).**
>
> Method        | FID ↓
> --------------|---------
> EditTransfer  | 4.908
> **Ours∩ET**   | **2.201**
> VisualCloze   | 7.218
> **Ours∩VC**   | **4.801**
>
> As shown in Table 4, compared with the in-context based method (without using RelationAdapter), our model also has a observable improvement in FID score.
>
> **Table 4: FID Results for Ablation Study (-S for Seen, -U for Unseen).**
>
> Method         | FID ↓
> ---------------|---------
> w/o RA -S      | 5.968
> **Ours -S**    | **5.191**
> w/o RA -U      | 5.571
> **Ours -U**    | **5.498**
>
>
> ## 3. The authors should discuss the effect of parameter $\alpha$ on generation. Further, the authors should provide an experiment to back their claim that SigLIP is a better alternative compared to VAE or CLIP for extracting visual prompt features from paired images.
>
> The attention fusion coefficient $\alpha$ controls the relative influence between the visual prompt attention produced by the RelationAdapter and the base Multi-Modal Attention (MMA) within the Diffusion Transformer. As described in Section 4.1 (Settings), we set $\alpha$ = 1 during training to maintain a balanced integration of visual guidance and generative consistency.
>
> Since $\alpha$ is part of the training configuration, we adopt the same value ($\alpha$ = 1) during inference to ensure consistency between training and deployment. Nevertheless, we further investigated the impact of varying $\alpha$ and report the results in Table 5. The evaluation confirms that maintaining $\alpha$ = 1—aligned with the training setup—yields the most stable and optimal generation performance.
>
> **Table 5: Effect of Adjusting Attention Fusion Coefficient $\alpha$ on Image Editing Quality (−S for Seen, −U for Unseen Tasks).**
>
> Method         | MSE ↓   | CLIP-I ↑  | FID ↓   | GPT-C ↑  | GPT-A ↑
> ---------------|---------|---------|---------|----------|---------
> $\alpha$ = 2 -S       | **0.044** | 0.827   | 5.564   | 3.855    | 3.536
> **$\alpha$ = 1 -S**   | **0.044** | **0.852** | **5.191** | **4.115**  | **4.258**
> $\alpha$ = 0.5 -S     | 0.050   | 0.832   | 5.895   | 4.099    | 3.591
> $\alpha$ = 2 -U       | 0.054   | 0.794   | 5.805   | 3.858    | 3.527
> **$\alpha$ = 1 -U**   | **0.053** | **0.812** | **5.498** | **4.211**   | **4.377**
> $\alpha$ = 0.5 -U     | 0.056   | 0.808   | 5.724   | 4.149  | 3.620
>
> Plus, our decision to use SigLIP in place of CLIP for visual prompt feature extraction is motivated by both prior empirical findings and architectural considerations. Specifically, CLIP has been widely observed to exhibit limited performance on fine-grained visual representation. Several studies replaced CLIP with SigLIP for generative and editing tasks, with ablations validating its effectiveness:
>
> Flux.1-Redux-dev [8] employs SigLIP as the image encoder in a diffusion framework for visual editing, citing improved visual prompt fidelity and detail preservation (Black Forest Labs). Generative Visual Instruction Tuning [9] replaces CLIP with SigLIP for paired visual-text instruction learning, demonstrating improved grounding and fidelity. MultiGen [10] leverages SigLIP to extract reference features for multi-modal prompt-based image generation, showing stronger visual-semantic alignment compared to CLIP. Insert Anything [11] integrates SigLIP into a DiT-based pipeline for object insertion, enabling better semantic conditioning from reference images. Calligrapher [12] uses SigLIP for personalized text-image customization and finds it superior in capturing local stylistic variations. UniToken [13] unifies visual understanding and generation using SigLIP to encode dense visual tokens in lieu of CLIP, enhancing both reconstruction and cross-modal alignment.
>
> ## 4. The total number of trainable parameters are 1.5B which is relatively high.
>
> We conducted experiments with a lower-rank setting (rank=16), which resulted in a significantly smaller parameter footprint of approximately 1.25B. As shown in Table 6, our method remains effective even under this lightweight configuration.
>
> **Table 6: Comparison Between LoRA Rank=16 and Our Original Experimental Setting (−S for Seen, −U for Unseen).**
>
> Method              | MSE ↓   | CLIP-I ↑  | FID ↓   | GPT-C ↑  | GPT-A ↑
> --------------------|---------|---------|---------|----------|---------
> Ours -S (Rank=16)   | 0.048   | 0.828   | 5.757   | 4.035    | 3.607
> **Ours -S**         | **0.044** | **0.852** | **5.191** | **4.145**  | **4.219**
> Ours -U (Rank=16)   | 0.064   | 0.792   | 5.924   | 4.026    | 3.505
> **Ours -U**         | **0.053** | **0.812** | **5.498** | **4.195**  | **4.239**
>
> In practice, we observe that the optimal LoRA rank correlates with dataset size and diversity. Given the scale of our Relation252K dataset, we opted for a higher rank (128) to fully exploit its richness.
>
>
> [1] Lan Chen, Qi Mao, Yuchao Gu, and Mike Zheng Shou. Edit transfer: Learning image editing via vision in-context relations.
>
> [2] Zhong-Yu Li, Ruoyi Du, Juncheng Yan, Le Zhuo, Zhen Li, Peng Gao, Zhanyu Ma, and Ming-Ming Cheng. Visualcloze: A universal image generation framework via visual in-context learning.
>
> [3] Amir Hertz, Ron Mokady, Jay Tenenbaum, Kfir Aberman, Yael Pritch, and Daniel Cohen-Or. Prompt-to-prompt image editing with cross attention control. In ICLR, 2023.
>
> [4] Jiangshan Wang, Junfu Pu, Zhongang Qi, Jiayi Guo, Yue Ma, Nisha Huang, Yuxin Chen, Xiu Li, and Ying Shan. Taming rectified flow for inversion and editing.
>
> [5] Xi Chen, Yutong Feng, Mengting Chen, Yiyang Wang,Shilong Zhang, Yu Liu, Yujun Shen, and Hengshuang Zhao. Zero-shot image editing with reference imitation. In NeurIPS, 2025.
>
> [6] Zhenxiong Tan, Songhua Liu, Xingyi Yang, Qiaochu Xue, and Xinchao Wang. Ominicontrol: Minimal and universal control for diffusion transformer.
>
> [7] Chen, Xi, et al. "Unireal: Universal image generation and editing via learning real-world dynamics." In CVPR, 2025.
>
> [8] Black Forest Labs. (2024).
>
> [9] Hernandez, J., Villegas, R., and Ordonez, V. Generative Visual Instruction Tuning. arXiv 2024.
>
> [10] Wu, Z.-F., Huang, L., Wang, W., Wei, Y., and Liu, Y. MultiGen: Zero-shot Image Generation from Multi-modal Prompts. In ECCV, 2024.
>
> [11] Song, W., Jiang, H., Yang, Z., Quan, R., and Yang, Y. Insert Anything: Image Insertion via In-Context Editing in DiT. arXiv 2025.
>
> [12] Ma, Y., Bai, Q., Ouyang, H., Cheng, K. L., Wang, Q., Liu, H., … Chen, Q. Calligrapher: Freestyle Text Image Customization. arXiv 2025.
>
> [13] Jiao, Y., Qiu, H., Jie, Z., Chen, S., Chen, J., Ma, L., and Jiang, Y.-G. UniToken: Harmonizing Multimodal Understanding and Generation through Unified Visual Encoding. arXiv 2025.

---

> ### Comment · Reviewer_s3JJ · 2025-08-06
>
> Thanks to the authors for their response. While my concerns regarding quality of generated images, comparison with baselines under same LORA rank, and motivation for the use of SigLIP have been resolved, my concerns around limited baselines, ablation for parameter  $\alpha$, and large number of trainable parameters still remain. 1) Example pair used in the proposed method can be interpreted as visual conditions and text prompt can be used to guide UniReal[7] to generate image from source image following visual relation of the pair (for example, "Generate RES1 from IMG3 by applying the visual relation of IMG2 from IMG1"). 2) It is expected that the  $\alpha$ used in training would be the optimal value during inference. However, effect of varying  $\alpha$ in the training is unclear and  $\alpha=1$ might not be the optimal value for best performance. 3) Rank 16 does not lead to a significant reduction in the parameter count which reduces from 1.5B to around 1.25B. Hence, I will keep my score.

---

> > ### Author Response · Authors · 2025-08-06
> >
> > We thank the reviewer for the valuable feedback on improving this paper! Please find below our response to the reviewer’s questions.
> >
> > ## 1. Example pair used in the proposed method can be interpreted as visual conditions and text prompt can be used to guide UniReal[1] to generate image from source image following visual relation of the pair.
> >
> > Regarding the potential applicability of UniReal [1] for our setting, we are also very interested in evaluating UniReal's capabilities in this context. However, as of the current date, UniReal’s official repository only includes the dataset construction code and lacks both training and inference implementations, making a direct empirical comparison infeasible.
> >
> > In fact, UniReal's approach of concatenating multiple images after VAE encoding is quite similar to in-context learning-based methods. In our paper, we also acknowledge that such concatenation followed by bidirectional attention is indeed effective. However, our proposed method builds upon this line of research by introducing a key innovation: we decouple the condition image tokens from the original image tokens and fuse them via a cross-attention mechanism rather than simple concatenation. This design not only improves editing consistency and accuracy (Table 1), but also reduces GPU memory consumption and accelerates both training and inference (Table 2).
> >
> > **Table 1: Ablation Study on the Effectiveness of the RelationAdapter (RA) in Seen and Unseen Tasks.**
> >
> > Method      | MSE ↓    | CLIP-I ↑ | GPT-C ↑ | GPT-A ↑
> > ------------|----------|----------|---------|---------
> > w/o RA -S   | 0.055    | 0.787    | 3.909   | 3.597
> > **Ours -S** | **0.044**| **0.852**| **4.079**| **4.106**
> > w/o RA -U   | 0.061    | 0.778    | 3.840   | 3.566
> > **Ours -U** | **0.053**| **0.812**| **4.187**| **4.173**
> >
> > **Table 2: Comparison of Computational Efficiency Between In-Context Baseline and RelationAdapter (Ours). - Inference on 1024×1024 Images**
> >
> > Method                  | GPU Memory (GB) ↓ | Training Time (hrs) ↓ | Inference Time (s) ↓ | Training Speed-Up ↑ | Inference Speed-Up ↑
> > ------------------------|-------------------|------------------------|-----------------------|----------------------|------------------------
> > In-Context Baseline     | 77                | 51.5                   | 13.0+                 | —                    | —
> > **RelationAdapter (Ours)**  | **74**              | **48**                   | **< 9.0**               | **↑6.8%**              | **↑30.8%**
> >
> > ## 2. It is expected that the $\alpha$ used in training would be the optimal value during inference. However, effect of varying $\alpha$ in the training is unclear and $\alpha = 1$ might not be the optimal value for best performance.
> >
> > Thank you for your insightful comment. The current setting of the attention fusion coefficient $\alpha = 1$ was chosen based on empirical findings and in line with prior works such as IP-Adapter [2], Ada-Adapter [3], and VMix [4], all of which adopt a default fusion weight of 1 during attention injection.
> >
> > In our case, the visual prompt tokens have a length of 256, which is aligned with the length of the text prompt tokens. Setting $\alpha = 1$ ensures that visual and textual prompts are treated equally during fusion, which we believe is both intuitive and semantically reasonable.
> >
> > That said, we appreciate your suggestion and agree that exploring the effect of varying $\alpha$ during training could provide valuable insights. While full experimentation requires significant training time, we will include a discussion with different values of $\alpha$ in the appendix of the revised version.
> >
> > ## 3. Rank 16 does not lead to a significant reduction in the parameter count which reduces from 1.5B to around 1.25B.
> >
> > We would like to clarify that the 1.25B parameter figure refers to the total number of trainable parameters, including the newly introduced linear projection layers. When isolating the LoRA parameters alone, the reduction from rank 128 to rank 16 is indeed substantial: the number of trainable parameters decreases from **358.6M** to **44.8M**, representing an **87.5%** reduction in trainable LoRA parameters.
> >
> > Moreover, our method indeed offers notable improvements in both training and inference efficiency compared to the in-context learning baseline, as shown in Table 2.
> >
> > References:
> >
> > [1] Chen, Xi, et al. "Unireal: Universal image generation and editing via learning real-world dynamics." In CVPR, 2025.
> >
> > [2] Ye, Hu, et al. "Ip-adapter: Text compatible image prompt adapter for text-to-image diffusion models." arXiv 2023.
> >
> > [3] Liu, Jia, et al. "Ada-adapter: Fast few-shot style personlization of diffusion model with pre-trained image encoder." arXiv 2024
> >
> > [4] Wu, Shaojin, et al. "VMix: Improving Text-to-Image Diffusion Model with Cross-Attention Mixing Control." arXiv 2024.
> >
> > Again, we thank the reviewer for the valuable feedback. Please let us know if there are any other questions or suggestions.
> >
> > Best,
> >
> > Authors

---

### Official Review · Reviewer_Bkni · 2025-07-01

**Clarity:** 3
**Significance:** 2
**Originality:** 2
**Rating:** 3
**Confidence:** 4

**Summary:**

The paper proposes RelationAdapter, a lightweight module for DiT-based models that enables effective transfer of visual editing intent from source-target pairs, and introduces a large-scale dataset, Relation252K, to benchmark generalizable image editing. Experiments demonstrate improved generation quality and editing performance.

**Questions:**

1. In the RelationAdapter section, after revisiting visual prompt integration, the authors criticize the current FLUX-1 Redux by stating: 'visual feature embeddings are typically much longer than textual prompts, which can weaken or even nullify text-based guidance.' However, based on the presented results, the primary editing guidance appears to originate from source-target image pairs rather than text-based guidance. In fact, most demonstrated cases don't even include text prompts. Does this imply that text guidance is fundamentally unimportant for this task? Consequently, might FLUX-1 Redux actually not be problematic?
2. In the RelationAdapter section, the authors should clarify why RA Attention requires the Q from MM Attention and what specific information it encodes. Additionally, does introducing the RA Attention module double the computational cost of the diffusion process?
3. Most demos in the supplementary materials appear to focus on human subjects. Does this indicate stronger performance on human images? The authors should provide results across all four main task categories mentioned.

**Ethical Concerns:**

["NO or VERY MINOR ethics concerns only"]

**Final Justification:**

3: Borderline reject.

**Limitations:**

yes

**Quality:**

3

**Strengths And Weaknesses:**

Strengths:
1. Position Encoding Cloning appears to offer a promising solution for pixel-level misalignment between source and target images.
2. The introduction of a new image editing dataset with full open-source commitment constitutes a valuable contribution to the image editing community.

Weakness：
1. In the RelationAdapter section, after revisiting visual prompt integration, the authors criticize the current FLUX-1 Redux by stating: 'visual feature embeddings are typically much longer than textual prompts, which can weaken or even nullify text-based guidance.' However, based on the presented results, the primary editing guidance appears to originate from source-target image pairs rather than text-based guidance. In fact, most demonstrated cases don't even include text prompts. Does this imply that text guidance is fundamentally unimportant for this task? Consequently, might FLUX-1 Redux actually not be problematic?
2. In the RelationAdapter section, the authors should clarify why RA Attention requires the Q from MM Attention and what specific information it encodes. Additionally, does introducing the RA Attention module double the computational cost of the diffusion process?
3. Most demos in the supplementary materials appear to focus on human subjects. Does this indicate stronger performance on human images? The authors should provide results across all four main task categories mentioned.
4. The comparative experiments include only two baselines, which is limited. The ablation studies should further dissect individual components within each module.

---

> ### Author Rebuttal · Authors · 2025-07-31
>
> ## 1. In the RelationAdapter section, after revisiting visual prompt integration, the authors criticize the current FLUX-1 Redux by stating: 'visual feature embeddings are typically much longer than textual prompts, which can weaken or even nullify text-based guidance.' However, based on the presented results, the primary editing guidance appears to originate from source-target image pairs rather than text-based guidance. In fact, most demonstrated cases don't even include text prompts. Does this imply that text guidance is fundamentally unimportant for this task? Consequently, might FLUX-1 Redux actually not be problematic?
>
> First of all, we sincerely thank the reviewer for this insightful question. Below is our response:
> Redux was originally positioned as a model for generating diverse image variants, which is stated on the homepage of the black-forest-labs/flux GitHub project. Image variants here refer to outputs that tend to closely resemble the input image. However, it has been widely observed by users that Redux fails to properly respond to text prompts—when a prompt is provided alongside an input image, the prompt is often ignored. This limitation has led to the development of several community-driven extensions aiming to improve prompt control within Redux, such as ComfyUIAdvancedRefluxControl.
>
> As mentioned in our paper, after analyzing the Redux codebase, we found the root cause lies in its prompt integration strategy: the text prompt is encoded using a T5 encoder, and the resulting text token sequence is simply concatenated with the image token sequence. Due to this, the textual input is significantly shorter than the visual input and easily overwhelmed. For example, a single conditioning image of size 768×1360 produces approximately (768 / 16) × (1360 / 16) = 4080 image tokens, whereas the number of text tokens is typically capped at 256 << 4080.
>
> This imbalance becomes even more severe in our source-target image pair setting, where three conditioning images(an original image, a source image showing the pre-edit state, and a target image showing the post-edit state) are used, leading to 4080 × 3 = 12,240 image tokens versus 256 text tokens. To address this, we propose the RelationAdapter, which encodes source and target images into two lightweight 128-token sequences. This not only reduces image token length drastically but also brings it into balance with the text sequence, resulting in a combined length of 128×2 = 256—matching the text input. Surprisingly, this compression of visual tokens not only reduces computational cost, but also improves editing performance compared to prior in-context learning approaches that directly concatenate high-dimensional image tokens.
>
> In order to demonstrate the impact of prompts on the RelationAdapter's editing results, we conducted additional ablation experiments and generated results using input without prompts. The evaluation results are presented in Table 1.
>
> **Table 1: Quantitative Comparison of Methods with and without Prompts (-S for Seen, -U for Unseen).**
>
> Method          | MSE ↓   | CLIP-I ↑  | GPT-C ↑  | GPT-A ↑
> ----------------|---------|---------|---------|---------
> w/o Prompt -S   | 0.065   | 0.777   | **4.272**   | 3.413
> **Ours -S**     | **0.044** | **0.852** | 3.997 | **4.353**
> w/o Prompt -U   | 0.091   | 0.745   | **4.391**   | 3.222
> **Ours -U**     | **0.053** | **0.812** | 3.973 | **4.293**
>
> In the absence of textual guidance, the model tends to avoid modifying the original image, which explains the higher GPT-C scores (a metric indicating the consistency of the edited result with the original image). This is a normal phenomenon. In contrast, the performance of MSE, CLIP-I, and GPT-A is notably worse compared to the prompted scenario.
>
> As for the images in the supplementary material, they provide an intuitive comparison of generation results after being influenced by the text prompts. Thank you for your question — our released dataset will include all relevant content, including source-target image pairs and the corresponding editing instructions.
>
> ## 2. In the RelationAdapter section, the authors should clarify why RA Attention requires the Q from MM Attention and what specific information it encodes. Additionally, does introducing the RA Attention module double the computational cost of the diffusion process?
>
> 1. Why RA Attention requires the Q from MM Attention?
>
>     (1) Semantic alignment: By using DiT’s backbone queries, which are conditioned on the source image and textual cues, the RA attention mechanism ensures that the visual cues adjust the attention mechanism based on the current generation context to avoid overfitting to the cue words.
>
>     (2) Efficient control: Sharing Q-values avoids an extra query projection layer and allows the RA attention mechanism to be interpreted as a soft modulation of the original attention mechanism, improving interpretability and stability.
>
> 2. What information does Q encode?
>
>     Essentially, the Q-value encodes the intent of the current generation (from the source and the text prompt), while the RA key/value encodes “how to edit” based on the example pairs. The RA module takes a sample pair and encodes the semantic shift and structural difference between the pre-edited and post-edited images. When the same Q-value is injected through the attention mechanism, it allows the model to adjust its generation based on “how similar content was edited before” while preserving the spatial context of the query.
>
> 3. Does introducing the RA Attention module double the computational cost of the diffusion process?
> The RA Attention module does not double the computational cost. RA attention introduces an additional lightweight KV attention layer in each DiT block, but it:
>
>     (1) Shares the existing Q computation with the original DiT self-attention mechanism;
>
>     (2) Uses a low-dimensional projection of SigLIP visual cue features (128D labels per image);
>
>     (3) Performs only one additional attention fusion step per block and uses a fixed scalar $\alpha$
>
>     When editing an image of size 1024×1024, the in-context based method requires over 13 seconds of inference, whereas our method only takes less than 9 seconds. This is because the complete concatenation of tokens greatly increases the computational time for attention, as we discussed in the second paragraph of the Introduction.
>
> ## 3. Most demos in the supplementary materials appear to focus on human subjects. Does this indicate stronger performance on human images? The authors should provide results across all four main task categories mentioned.
>
> We appreciate the reviewer's observation. Figure 5 (comparison results with baselines) does have a slightly higher proportion of human-centric examples. But this is unavoidable and is done for fair benchmarking—since Edit Transfer, one of our primary baselines, is specifically optimized for human appearance, pose, and outfit manipulation. Using non-human examples (e.g., animals, scenes) would risk an unfair comparison.
>
> Plus, each example in the supplementary corresponds to a distinct editing task, and collectively, they span all four main task categories described in the paper: 1. Low-level image processing (e.g., surfacenormal estimation, depth estimation) 2. Style transfer (e.g., crayon sketch, watercolor painting) 3. Image editing (e.g., object addition, cloth try-on) 4. Customized generation (e.g., Pokémon evolution, plush logo)
>
> At the same time, we show the evaluation results of the four task categories in Table 3 of the paper, where the human-preferred evaluation results (edit accuracy and edit consistency) of each evaluation result are around 4 (out of 5), demonstrating that in the four task categories, the ability of RelationAdapter can cover a variety of tasks of different complexity.
>
> **Table 3: Quantitative comparison of evaluation metrics (mean ± std) across four image generation tasks. Best results are shown in bold.**
>
> Tasks                         | MSE ↓        | CLIP-I ↑    | GPT-C ↑     | GPT-A ↑
> ------------------------------|--------------|-------------|-------------|-------------
> Low-Level (n=32)              | **0.028 ± 0.038**| **0.885 ± 0.067**| 3.943 ± 0.383| 3.822 ± 0.406
> Style Transfer (n=84)         | 0.051 ± 0.032| 0.846 ± 0.036| 4.077 ± 0.198| **4.246 ± 0.285**
> Image Editing (n=63)          | 0.031 ± 0.023| 0.861 ± 0.055| **4.173 ± 0.229**| 4.100 ± 0.400
> Customized Generation (n=39)  | 0.065 ± 0.048| 0.816 ± 0.073| 4.071 ± 0.224| 4.064 ± 0.313
>
> This ensures that the evaluation results cover the entire range of tasks, not just human-based editing. We will clarify this and annotate the category of each additional experimental result in the supplementary material in the final version.

---

> ### Author Response · Authors · 2025-08-06
> **Feel free not to respond**
>
> Thank you for your follow-up and for taking the time to review our rebuttal. We respect your decision to maintain the original score. While we understand that you remain unconvinced about the level of advancement demonstrated, we would like to take this opportunity to reiterate our key contributions.
>
> Specifically, we introduce **RelationAdapter**, the first DiT-based adapter module designed to extract edit intent and apply it to the original image. Compared to mainstream in-context learning-based approaches, our method offers clear advancements in training and inference efficiency(Table 1), as well as in editing accuracy and consistency (Table 2).
>
> In addition, we contribute an **In-Context Editor** architecture and a new dataset **Relation252K** covering 218 types of editing tasks, which we believe can serve as valuable resources for future research in image editing and intent understanding.
>
> **Table 1: Comparison of Computational Efficiency Between In-Context Baseline and RelationAdapter (Ours). - Inference on 1024×1024 Images**
>
> Method                  | GPU Memory (GB) ↓ | Training Time (hrs) ↓ | Inference Time (s) ↓ | Training Speed-Up ↑ | Inference Speed-Up ↑
> ------------------------|-------------------|------------------------|-----------------------|----------------------|------------------------
> In-Context Baseline     | 77                | 51.5                   | 13.0+                 | —                    | —
> **RelationAdapter (Ours)**  | **74**              | **48**                   | **< 9.0**               | **↑6.8%**              | **↑30.8%**
>
> **Table 2: Ablation Study on the Effectiveness of the RelationAdapter (RA) in Seen and Unseen Tasks.**
>
> Method      | MSE ↓    | CLIP-I ↑ | GPT-C ↑ | GPT-A ↑
> ------------|----------|----------|---------|---------
> w/o RA -S   | 0.055    | 0.787    | 3.909   | 3.597
> **Ours -S** | **0.044**| **0.852**| **4.079**| **4.106**
> w/o RA -U   | 0.061    | 0.778    | 3.840   | 3.566
> **Ours -U** | **0.053**| **0.812**| **4.187**| **4.173**
>
> Thank you again for your feedback. This response is solely intended to reaffirm our contributions — please feel free not to respond.
>
> Wishing you all the best,
>
> The Authors

---

### Official Review · Reviewer_ZkYa · 2025-07-02

**Clarity:** 3
**Significance:** 3
**Originality:** 3
**Rating:** 4
**Confidence:** 3

**Summary:**

The paper introduces RelationAdapter, a novel framework for visual prompt-based image editing inspired by in-context learning in large language models. RelationAdapter leverages source-target image pairs to extract and transfer editing intent to new query images, addressing limitations of existing methods that struggle with non-rigid transformations. The framework includes a lightweight RelationAdapter module to model visual relationships, an In-Context Editor for zero-shot editing with positional encoding cloning, and a large-scale dataset Relation252K (252K samples across 218 tasks) to evaluate generalization. Experiments demonstrate superior performance over baselines in metrics like MSE, CLIP-I, and GPT-based evaluations, showcasing improved editing accuracy and consistency.

**Questions:**

1. How is this work related to LayerCraft: Enhancing Text-to-Image Generation?

**Ethical Concerns:**

["Major Concern: Data privacy, copyright, and consent"]

**Limitations:**

The authors have discussed the limitations in their manuscript.

**Quality:**

3

**Strengths And Weaknesses:**

### Strengths
1.  The framework is novel at the time, with a good range of performance
2.  The In-Context Editor and positional encoding cloning enhance spatial alignment and fidelity, supporting robust zero-shot transfer to unseen tasks.
3.  Relation252K provides a diverse benchmark (218 tasks) for evaluating visual prompt-based editing, addressing gaps in task diversity and scale.
4. The ablation study is thorough.

### Weaknesses
1. There is a lack of analysis on potential failure case and rate?
2.  How well would the framework generalize to OOD examples?
3. Since the dataset contains human figures, I am not sure if this would require some ethics reviews (Perhaps the sources of data collection should be discussed).

---

> ### Author Rebuttal · Authors · 2025-07-31
>
> ## 1. How is this work related to LayerCraft: Enhancing Text-to-Image Generation?
>
> We thank the reviewer for pointing out the connection to LayerCraft [1]. We would like to clarify the relationship between RelationAdapter and LayerCraft in two key aspects:
>
> (1) Differences in the usage and source of conditional signals:
> LayerCraft employs a multi-agent pipeline (e.g., ChainArchitect) to decompose a complex text prompt "An apple on a table" into sub-tasks—such as “generate an apple,” “generate a table,” and “merge the two objects”—and then leverages a text-to-image model (FLUX) to generate intermediate images for each sub-component. These images, generated from decomposed text instructions, serve as structured visual conditions. In contrast, RelationAdapter operates under a visual example-pair setting, where no intermediate image generation or prompt decomposition is required. Instead, given a source-target image pair, the RelationAdapter directly extracts the editing intent (which may encompass style transfer, semantic changes, or structural modifications) and encodes this transformation into two compact 128-dimensional tokens, serving as lightweight relational prompts.
>
> (2) Differences in integration mechanisms with the diffusion backbone:
> LayerCraft incorporates multiple visual conditions by concatenating their latent features (from intermediate images) into the generation pipeline. To prevent interference, it applies foreground-background masking via bounding boxes. This approach resembles in-context learning in spirit but introduces significant overhead in token length and computation. In contrast, RelationAdapter avoids latent concatenation altogether. It reuses the MM-Attention module and performs cross-attention between the shared query from the DiT backbone and the editing-intent tokens extracted in step (1). As no additional token sequences are introduced, our method maintains constant memory consumption regardless of the input pair complexity. This design allows RelationAdapter to be a lightweight, plug-and-play adapter that efficiently applies learned visual relations without increasing token length or GPU burden, distinguishing it from existing in-context or composition-based frameworks.
>
> We appreciate the reviewer’s suggestion and will incorporate LayerCraft into the Related Work section in the final version to properly position our method within the broader landscape of text-to-image controllable generation.
>
> ## 2. There is a lack of analysis on potential failure case and rate?
>
> Thank you for your thoughtful comment. We would like to clarify that we have indeed provided an analysis of potential failure cases. As shown in Figure 11 in the Appendix, we include a qualitative examination of representative failure cases across four tasks: gesture editing, background pedestrian removal, document rectification, and image-to-sketch conversion. These examples reveal two primary limitations of our model: (1) challenges in fine-grained spatial alignment, and (2) challenges in restoring intricate textual details. We explicitly present and discuss these failure cases to identify areas where the model's performance can be improved.
>
> ## 3. How well would the framework generalize to OOD examples?
>
> We appreciate the reviewer’s question regarding the generalization of our framework to out-of-distribution (OOD) examples. To evaluate this, we specifically partitioned our test set to include 300 samples drawn from 10 tasks that were never encountered during training, as introduced in Section 4.2 (Benchmark). These tasks are a subset of the full test set (comprising 218 tasks in total), and we refer to them as unseen tasks. The results on these tasks are reported in the right half of Figures 5, 6, and 7.
>
> Our framework demonstrates promising generalization to these unseen tasks. As shown in Table 1, even without task-specific adaptation, the proposed RelationAdapter (RA) consistently improves performance over the baseline across all metrics—MSE, CLIP-I, GPT-C, and GPT-A—on both seen and unseen tasks. For example, on unseen tasks, incorporating RA reduces MSE from 0.061 to 0.053 and increases GPT-C from 3.840 to 4.187.
>
> **Table 1: Ablation Study on the Effectiveness of the RelationAdapter (RA) in Seen and Unseen Tasks (-S for Seen, -U for Unseen). The best results are denoted as Bold.**
>
> Method         | MSE ↓    | CLIP-I ↑ | GPT-C ↑ | GPT-A ↑
> ---------------|----------|----------|---------|---------
> w/o RA -S      | 0.055    | 0.787    | 3.909   | 3.597
> **Ours -S**    | **0.044** | **0.852** | **4.079** | **4.106**
> w/o RA -U      | 0.061    | 0.778    | 3.840   | 3.566
> **Ours -U**    | **0.053** | **0.812** | **4.187** | **4.173**
>
>
> These results demonstrate that our method, particularly with the RelationAdapter, retains strong performance even in truly unseen scenarios. We believe this supports the claim that our framework exhibits meaningful generalization to OOD settings.
>
> ## 4. Since the dataset contains human figures, I am not sure if this would require some ethics reviews (Perhaps the sources of data collection should be discussed).
>
> Thank you for raising this important concern. We would like to clarify that all the data used in our work are sourced from publicly available datasets that are widely adopted in the research community and fall within licenses that permit research and publication. Specifically, the image editing dataset was synthetically generated using GPT-4o and Midjourney, which ensures compliance with ethical and privacy standards. For the low-level vision datasets, all sources are properly cited in the paper. Since our work strictly adheres to the usage policies of the original datasets and does not involve any private data collection or interaction with human subjects, it does not require additional ethics approval.
>
> [1] Zhang Y, Li J, Tai Y W. Layercraft: Enhancing text-to-image generation with cot reasoning and layered object integration. arXiv 2025.

---

### Official Review · Reviewer_cG9Q · 2025-07-03

**Clarity:** 3
**Significance:** 3
**Originality:** 2
**Rating:** 5
**Confidence:** 5

**Summary:**

This paper proposes an exemplar-based image editing method capable of performing low-level editing, style transfer, customized generation, and other diverse tasks, with good generalization capability. The proposed RelationAdapter leverages a frozen image encoder to extract visual features that capture the relation between the source and target, injecting them into the diffusion process in an IP-Adapter fashion. Compared to existing in-context learning methods, this approach reduces GPU memory consumption and achieves a higher editing success rate. Strategies such as position encoding cloning and noise-free conditioning are applied to improve performance. The authors also construct a comprehensive dataset, Relation252K, for benchmarking. According to quantitative and qualitative results, RelationAdapter demonstrates superior performance on both seen and unseen tasks.

**Questions:**

1. The proposed dataset seems to be well-designed. However, a similar dataset Graph200K was proposed in VisualCloze. Why is this dataset not used for training and benchmarking?

2. The ablation study and comparison experiments should be extended, according to the reasons stated  in 'Weakness'.

3. What is the actual advantage (in terms of effectiveness and efficiency) of the proposed method compared to the in-context based variant? Please give further discussion of the claimed 'higher editing accuracy and consistency' including possible reasons behind this improvement, and add quantitative numbers of memory usage savings.

4. Position encoding cloning is clearly effective for pixel-wise aligned editing tasks. However, for other tasks like customized generation, is this strategy still reasonable? This problem seems to have been discussed in previous works [1], please give more explanations.

[1] OminiControl: Minimal and Universal Control for Diffusion Transformer

**Ethical Concerns:**

["NO or VERY MINOR ethics concerns only"]

**Final Justification:**

As the rebuttal addressed most of my concerns, I will keep a positive view of the paper and raise my score to accept.

**Limitations:**

yes

**Quality:**

3

**Strengths And Weaknesses:**

- Strengths:

1. The proposed RelationAdapter pipeline is reasonably designed and clearly explained. Compared with in-context learning-based methods, explicitly extracting visual features with pre-trained VLM does reduces memory consumption in diffusion models and may lead to better semantic understanding in the editing task.

2. The qualitative and quantitative results looks promising.

3. The proposed benchmark dataset iswell-constructed, with diverse editing tasks ranging from pixelwise-aligned depth estimation to free-formed customized generation. High-quality datasets are a valuable contribution to the community.


- Weakness:

1. The ablation study is not comprehensive enough. The effectiveness of key designs such as position encoding cloning and noise-free conditioning are not verified.

2. There are only two comparison methods. More comparisons are expected, including MidJourney (since the dataset is constructed with MidJourney, I am curious if the proposed method generate better results than the data source.)

3. Although the approach is sound, the advantage of using adapter-based approach over in-context-based approach is not well demonstrated. In Figure 6, the results is only slightly better than the in-context learning ablation variant. Also, the claimed reduction in memory consumption is not provided in numbers.

4. Some of the proposed strategies are debatable. For instance, position encoding cloning may not be suitable for tasks that are not pixel-wise aligned.

---

> ### Author Rebuttal · Authors · 2025-07-31
>
> ## 1. The proposed dataset seems to be well-designed. However, a similar dataset Graph200K was proposed in VisualCloze. Why is this dataset not used for training and benchmarking?
>
> We sincerely thank the reviewer for recognizing the design quality of our proposed dataset. While Graph200K introduced in VisualCloze [1] is indeed a valuable contribution, we would like to clarify that only a small fraction of its tasks align with the core setting of our work— relation-based image editing from example pairs. The vast majority of Graph200K samples are not directly compatible with the visual relation transfer paradigm we target.
>
> Moreover, incorporating Graph200K into our training pipeline would cause a significant imbalance in task distribution due to the limited number of applicable samples per task. In contrast, our Relation252K dataset guarantees at least 210 samples per task across 218 distinct editing tasks.
>
> Lastly, since there is partial task overlap between Relation252K and Graph200K, we opted not to include Graph200K in additional benchmarks to avoid redundant evaluation and ensure clarity in performance attribution.
>
> ## 2. The ablation study and comparison experiments should be extended, according to the reasons stated in 'Weakness'.
>
> We agree that broader baseline comparisons are beneficial. However, it is worth noting that RelationAdapter uniquely requires a complete 'before-after' pair as context—a formulation shared only with Edit Transfer [10] and VisualCloze [1]. We will clarify this distinction in the final version. Another reviewer referred to several related studies, which we further differentiate from RelationAdapter below:
>
> Prompt-to-Prompt [2] and RF-Edit [3] operate entirely in the text-driven editing paradigm. They do not utilize any visual examples and thus lack the capacity to learn visual editing transformations from exemplar pairs. Zero-shot Image Editing [4] and OminiControl [5] focus primarily on reference-conditioned generation, where an auxiliary image—typically a Canny edge map, depth map, segmentation mask, or color scribble—is used to provide fine-grained control signals during image generation. The core idea is to apply a predefined image condition to guide image generation, rather than to learn and transfer a transformation between image pairs. UniReal [6] addresses a different task setting—combining multiple conditioning images (e.g., “placing the cat from IMG1 onto the bed in IMG2, guided by a mask in IMG3”).
>
> In the 'weakness' section, you also suggested using Midjourney as a comparative baseline. However, it is important to clarify that Midjourney does not support generation conditioned on three input images. Its interface only differentiates between a single character reference (cref) and a single style reference (sref). The exemplar pair in RelationAdapter serves as a visual demonstration of edit intent, which is distinct from either character or style references in Midjourney.
>
> Nevertheless, we conducted experiments to evaluate Midjourney's capabilities by removing the pre-edited example image input. To maintain consistency with the original input, we used the original image as the cref and the edited example image as the sref. It is worth noting that, for fairness, we chose style transfer tasks that RelationAdapter was not trained on for comparison with Midjourney. The text prompt used in Midjourney is as follows:
>
> $text\ prompt\ --cref\ <source\ image>\ --sref\ <edited\ image\ in\ the\ editing\ example\ pair>\ --cw\ 90\ --sw\ 70\ --v\ 6.1$
>
> Table 1 compares the performance of our proposed method, RelationAdapter, with Midjourney (MJ) on unseen style transfer tasks. Despite not having seen these specific tasks during training, our model demonstrates consistently superior results across all evaluation metrics.
>
> **Table 1: Comparison of RelationAdapter and MJ (Midjourney) on Unseen Style Transfer Tasks. The best results are denoted as Bold.**
>
> Method      | MSE ↓   | CLIP-I ↑  | GPT-C ↑ | GPT-A ↑
> ------------|---------|---------|---------|---------
> MJ          | 0.107| 0.681  | 3.285  | 3.200
> **Ours**    | **0.062** | **0.774** | **4.203** | **4.278**
>
> Plus, we acknowledge that our current ablation study does not separately isolate the effects of position encoding cloning and noise-free conditioning. These settings are already backed by both theoretical intuition and empirical precedent in recent literature such as LayerCraft [7], OminiControl [5], EasyControl [8] and Kontext [9], reinforcing their empirical effectiveness as standard paradigms in Transformer-based diffusion models.
>
> ## 3. What is the actual advantage (in terms of effectiveness and efficiency) of the proposed method compared to the in-context based variant? Please give further discussion of the claimed 'higher editing accuracy and consistency' including possible reasons behind this improvement, and add quantitative numbers of memory usage savings.
>
> Regarding the actual advantage over the in-context based variant in terms of effectiveness and efficiency, we provide the following comparative analysis:
>
> Training Efficiency: The in-context based method required approximately 77 GB of GPU memory and a total of 51.5 hours of training time. In contrast, our proposed RelationAdapter used around 74 GB of GPU memory and completed training in approximately 48 hours. This represents a memory saving of ~3 GB and a 6.8\% reduction in training time.
>
> Inference Efficiency: For editing a single image of resolution 1024×1024, the in-context based method requires over 13 seconds of inference, whereas our method only takes less than 9 seconds, achieving a 30.8\% speed-up in inference.
>
> These improvements arise from a crucial architectural difference: as mentioned in the second paragraph of the introduction, the in-context method, which employs complete token concatenation, increases the computational load on the attention mechanism, leading to slower inference speeds. In contrast, RelationAdapter utilizes a decoupled attention mechanism, where computations are performed separately and subsequently fused, resulting in more efficient processing.
>
> Regarding higher editing accuracy and consistency, we still attribute these improvements to the decoupled attention mechanism. This mechanism helps avoid contamination between the original image features and the exemplar image pair features, resulting in more targeted and consistent edits. Both quantitative evaluations (e.g., Table 2) and qualitative results (e.g., Figure 6) support this rationale, showing that RelationAdapter consistently outperforms the in-context variant. We will emphasize these improvements in the final version of the paper, as we believe they clearly illustrate the concrete benefits and design motivations behind our approach.
>
> ## 4. Position encoding cloning is clearly effective for pixel-wise aligned editing tasks. However, for other tasks like customized generation, is this strategy still reasonable? This problem seems to have been discussed in previous works [5], please give more explanations.
>
> PEC was designed to enforce spatial consistency without requiring strict spatial correspondence. Instead, it offers a soft spatial prior to guide attention, allowing for the editing of semantically aligned yet structurally flexible regions.
>
> OminiControl [5] does mention using different positional strategies for pixel-aligned and non-aligned tasks. However, we observe that the loss plots for shared position indexing and shifting position indexing—shown in Figure 3(b) of their paper—converge to the same value for both strategies as the number of training steps approaches 1000.
>
> Since our goal is to build an image editing model capable of handling a wide range of tasks, PEC provides a soft spatial prior to guide the model in learning flexible semantic alignment. After careful consideration, we decided not to introduce a fixed offset. Our experimental results demonstrate that PEC is effective even for complex or non-aligned tasks such as background replacement and composition editing. We will clarify this in the final version to emphasize the role of PEC as a non-strict guidance mechanism.
>
>
>
> [1] Li Z Y, Du R, Yan J, et al. VisualCloze: A Universal Image Generation Framework via Visual In-Context Learning. arXiv 2025.
>
> [2] Amir Hertz, Ron Mokady, Jay Tenenbaum, Kfir Aberman, Yael Pritch, and Daniel Cohen-Or. Prompt-to-prompt image editing with cross attention control. In ICLR, 2023.
>
> [3] Jiangshan Wang, Junfu Pu, Zhongang Qi, Jiayi Guo, Yue Ma, Nisha Huang, Yuxin Chen, Xiu Li, and Ying Shan. Taming rectified flow for inversion and editing.
>
> [4] Xi Chen, Yutong Feng, Mengting Chen, Yiyang Wang,Shilong Zhang, Yu Liu, Yujun Shen, and Hengshuang Zhao. Zero-shot image editing with reference imitation. In NeurIPS, 2025.
>
> [5] Zhenxiong Tan, Songhua Liu, Xingyi Yang, Qiaochu Xue, and Xinchao Wang. Ominicontrol: Minimal and universal control for diffusion transformer.
>
> [6] Chen, Xi, et al. "Unireal: Universal image generation and editing via learning real-world dynamics." Proceedings of the Computer Vision and Pattern Recognition Conference. 2025.
>
> [7] Zhang Y, Li J, Tai Y W. Layercraft: Enhancing text-to-image generation with cot reasoning and layered object integration. arXiv 2025.
>
> [8] Zhang, Yuxuan, et al. Easycontrol: Adding efficient and flexible control for diffusion transformer. arXiv 2025.
>
> [9] Batifol, Stephen, et al. FLUX. 1 Kontext: Flow Matching for In-Context Image Generation and Editing in Latent Space. arXiv 2025.
>
> [10] Lan Chen, Qi Mao, Yuchao Gu, and Mike Zheng Shou. Edit transfer: Learning image editing via vision in-context relations.

---

> > ### Comment · Reviewer_cG9Q · 2025-08-05
> >
> > Thanks to the authors for their reply. As the rebuttal addressed most of my concerns, I will keep a positive view of the paper and may raise my score to accept after the discussion period. I look forward to seeing other reviewers' opinions.

---

> > > ### Author Response · Authors · 2025-08-06
> > >
> > > Thank you for taking the time to review our rebuttal. We're pleased that our responses addressed your concerns, and we sincerely appreciate your willingness to reconsider the score.
> > >
> > > Best,
> > >
> > > Authors

---

### Note · Authors · 2025-08-12

We thank all reviewers, ACs, and PCs for their valuable comments and discussions, which greatly improved our work through new experiments, clarifications, and analyses.

Our paper addresses visual relation transfer—learning editing transformations from example pairs—and makes three main contributions:

**Method** – We introduce **RelationAdapter**, a lightweight add-on for Diffusion Transformers that learns edits from example pairs. By encoding source–target examples into two 128-length token sequences and integrating them via **decoupled cross-attention**, our method avoid the computation and semantic noise of in-context concatenation, delivering more accurate, more consistent results with lower memory and latency. We also design an **In-Context Editor** framework that introduces **Position Encoding Cloning (PEC)** and a **noise-free conditioning** paradigm; together they ensure pixel-level spatial alignment and clean conditional signals throughout denoising, yielding more stable, controllable edits with fewer artifacts and stronger generalization to unseen tasks.

**Dataset** – We release **Relation252K**, a large, well-structured benchmark covering **218** editing tasks with ample samples, designed for fair and comprehensive evaluation of relation-based editing. We also propose an **automatic dataset construction pipeline** that generates diverse, high-quality source–target pairs while avoiding task mismatches and redundancy, enabling scalable and reproducible benchmark creation.

**Experiments & Analysis** – We achieve **state-of-the-art (SOTA)** results, with consistent improvements over strong in-context baselines and Midjourney across standard metrics on both seen and unseen tasks. Following reviewer feedback, we **added new experiments**. Results with **lower-rank LoRA (rank = 16)** demonstrate that our approach remains effective under reduced capacity, highlighting its **scalability** to lighter configurations. We further analyze the effect of the **attention fusion coefficient** as well as the influence of **text prompts** on editing quality. We also provide thorough analyses and will **open-source all code and data** for reproducibility.

In summary, this work advances relation-based image editing in method design, system architecture, and benchmark construction, delivering both accuracy and efficiency gains with a publicly available large-scale dataset.

---

### Decision · Program_Chairs · 2025-09-17

**Decision:**

Accept (poster)

**Comment:**

1x Accept, 1x Borderline Accept, 2x Borderline Reject: The paper proposes an in-context learning method for image editing, guided by a single "before-and-after" example. Reviewers acknowledge several strengths of the paper, including technical soundness, clear presentation, and a comprehensive dataset with 218 diverse editing tasks. The weaknesses noted by reviewers include limited baseline comparisons (only two methods) and insufficient ablation studies. In the rebuttal, the authors provided extensive new experiments including FID scores, Midjourney comparisons, ablations, and lower-rank LoRA tests addressing concerns about the 1.5B parameter count. They also clarified that suggested baselines like Prompt-to-Prompt are text-only methods that don't use visual examples, while others like OminiControl use auxiliary control images (depth maps, edges) rather than learning transformations from example pairs. Regarding UniReal, they noted its repository only includes dataset construction code without training or inference implementations. Two reviewers found the rebuttal convincing (cG9Q changed to Accept, ZkYa's concerns were addressed), while Bkni provided minimal engagement and s3JJ maintained baseline concerns about UniReal despite partial resolution.

The AC recommends acceptance. The paper presents a technically sound approach with compelling empirical results. The authors' comprehensive rebuttal with new experiments adequately addresses the primary concerns and support acceptance. While the 2-2 split reflects differing views, Bkni provided minimal engagement and the inability to compare to s3JJ's suggested baseline UniReal due to lack of available implementation makes this concern impractical.

Authors are advised to revise based on reviewers' comments to make the final version more solid.